# Casein kinase 1 family proteins promote Slimb-dependent Expanded degradation

**Alexander D Fulford[1,2†], Maxine V Holder[3†], David Frith[4], Ambrosius P Snijders[4], Nicolas Tapon[3]\*, Paulo S Ribeiro[1]\***

[1]Centre for Tumour Biology, Barts Cancer Institute, Queen Mary University of London, London, United Kingdom; [2]Department of Developmental Biology, Washington University School of Medicine, St. Louis, United States; [3]Apoptosis and Proliferation Control Laboratory, The Francis Crick Institute, London, United Kingdom; [4]Proteomics, The Francis Crick Institute, London, United Kingdom

**Abstract** Hippo signalling integrates diverse stimuli related to epithelial architecture to regulate tissue growth and cell fate decisions. The Hippo kinase cascade represses the growth-promoting transcription co-activator Yorkie. The FERM protein Expanded is one of the main upstream Hippo signalling regulators in *Drosophila* as it promotes Hippo kinase signalling and directly inhibits Yorkie. To fulfil its function, Expanded is recruited to the plasma membrane by the polarity protein Crumbs. However, Crumbs-mediated recruitment also promotes Expanded turnover via a phosphodegron-mediated interaction with a Slimb/β-TrCP SCF E3 ligase complex. Here, we show that the Casein Kinase 1 (CKI) family is required for Expanded phosphorylation. CKI expression promotes Expanded phosphorylation and interaction with Slimb/β-TrCP. Conversely, CKI depletion in S2 cells impairs Expanded degradation downstream of Crumbs. In wing imaginal discs, CKI loss leads to elevated Expanded and Crumbs levels. Thus, phospho-dependent Expanded turnover ensures a tight coupling of Hippo pathway activity to epithelial architecture.

DOI: https://doi.org/10.7554/eLife.46592.001

**\*For correspondence:**
nic.tapon@crick.ac.uk (NT);
p.baptista-ribeiro@qmul.ac.uk
(PSR)

[†]These authors contributed
equally to this work

**Competing interests:** The authors declare that no competing interests exist.

## Introduction

The maintenance of epithelial tissue architecture through cell-cell and cell-extracellular matrix contacts, as well as cell polarity, is essential for organ function and size control (*Genevet and Tapon, 2011*; *Low et al., 2014*). The evolutionarily conserved Hippo (Hpo) pathway, a key signalling module that senses and responds to epithelial organisation, has emerged as a critical regulator of growth and epithelial integrity (*Genevet and Tapon, 2011*; *Schroeder and Halder, 2012*; *Yu and Guan, 2013*). At the core of Hpo signalling is a kinase cascade comprising Hpo and Warts (Wts), which promote the phosphorylation and inactivation of the pro-growth transcriptional co-activator Yorkie (Yki, YAP in mammals), thereby repressing tissue growth (*Yu and Guan, 2013*; *Fulford et al., 2018*; *Hong and Guan, 2012*). When Hpo signalling is inactive, Yki/YAP is able to enter the nucleus, associate with its transcription factor partner Scalloped (TEAD1-4 in mammals) and promote the expression of cell cycle regulators and apoptosis inhibitors, among others (*Genevet and Tapon, 2011*; *Yu and Guan, 2013*; *Hong and Guan, 2012*). To ensure that epithelial homeostasis is maintained, Yki/YAP also control the expression of Hpo pathway upstream regulators that dampen Yki/YAP activity as part of a negative feedback mechanism (*Genevet and Tapon, 2011*; *Halder and Johnson, 2011*).

Yki/YAP activity responds to epithelial organisation through the actin cytoskeleton (*Gaspar and Tapon, 2014*), basolateral polarity determinants (*Chen et al., 2012*; *Cordenonsi et al., 2011*; *Grzeschik et al., 2010*), adherens junction components, such as α-catenin (*Schlegelmilch et al., 2011*), and apical polarity proteins, such as Crumbs (Crb) (*Grzeschik et al., 2010*; *Chen et al., 2010*;

*Hafezi et al., 2012*; *Ling et al., 2010*; *Robinson et al., 2010*; *Varelas et al., 2010*). While the vast majority of these inputs act through Hpo and/or Wts, some engage core kinase cascade-independent signalling (*Genevet and Tapon, 2011*; *Yu and Guan, 2013*). The FERM domain protein Expanded (Ex) is a key regulator of Yki function (*Hamaratoglu et al., 2006*), which inhibits tissue growth using both Hpo-dependent and -independent mechanisms, in a manner analogous to mammalian Angiomotins (*Moleirinho et al., 2014*). Ex is an upstream activator of the Hpo core kinase cassette that forms a complex with Kibra and Merlin (*Baumgartner et al., 2010*; *Genevet et al., 2010*; *McCartney et al., 2000*; *Yu et al., 2010*). Ex has been proposed to promote core kinase activity by bridging Hpo association with the upstream kinase Tao-1 together with Schip1 (*Genevet and Tapon, 2011*; *Chung et al., 2016*), and by recruiting Wts to the apical membrane where it can be activated by Hpo (*Sun et al., 2015*). Ex also restrains tissue growth in a phosphorylation-independent manner by tethering Yki at the apical membrane via a direct PPxY:WW domain-mediated interaction (*Badouel et al., 2009*; *Oh et al., 2009*), a process regulated by Ack-dependent tyrosine phosphorylation of Ex (*Hu et al., 2016*). Moreover, Ex has also been linked with F-actin-mediated regulation of Yki function, by antagonising the action of Zyxin (*Gaspar et al., 2015*). Interestingly, *ex* is a Yki target gene and, therefore, it is a prime candidate to mediate the Hpo pathway feedback regulation that controls tissue homeostasis (*Genevet and Tapon, 2011*; *Halder and Johnson, 2011*).

Ex is a key link between epithelial polarity and Hpo signalling. This is controlled by the transmembrane protein Crb, which in addition to its recognised role in polarity, also regulates tissue growth by modulating the Notch and Hpo pathways (*Grzeschik et al., 2010*; *Chen et al., 2010*; *Hafezi et al., 2012*; *Ling et al., 2010*; *Robinson et al., 2010*; *Richardson and Pichaud, 2010*). Via its FERM-binding motif (FBM), Crb recruits Ex to the apical membrane, where it can promote inhibition of Yki (*Genevet and Tapon, 2011*; *Chen et al., 2010*; *Hafezi et al., 2012*; *Ling et al., 2010*; *Robinson et al., 2010*; *Su et al., 2017*). However, besides activating Hpo signalling through Ex, Crb also stimulates Ex phosphorylation and turnover of Ex protein (*Genevet and Tapon, 2011*; *Grzeschik et al., 2010*; *Chen et al., 2010*; *Ling et al., 2010*; *Robinson et al., 2010*; *Ribeiro et al., 2014*; *Laprise, 2011*). We have previously shown that ubiquitylation and degradation of Ex downstream of Crb is mediated by the phospho-dependent SCF$^{Slimb/\beta\text{-}TrCP}$ E3 ligase complex, which is also thought to regulate Ex levels independently of Crb (*Ribeiro et al., 2014*; *Zhang et al., 2015*). However, the identity of the kinase(s) that promote Ex degradation downstream of Crb is currently unknown.

Here, we identify the Casein Kinase 1 (CKI) family of protein kinases as regulators of Ex stability that function downstream of the polarity protein Crb. Depletion of CKI kinases suppresses Crb-induced Ex phosphorylation, ubiquitylation and degradation. Interestingly, CKI kinases regulate Ex in a partially redundant manner, which suggests that regulation of Ex stability is a key step in the regulation of Yki function and the maintenance of tissue homeostasis.

## Results

### Crb promotes interaction of Ex with Casein kinase 1 family proteins

We have previously shown that Crb regulates Ex protein stability in a β-TrCP-dependent manner (*Ribeiro et al., 2014*). The Ex:Slimb (Slmb, *Drosophila* β-TrCP) interaction is mediated by a β-TrCP consensus sequence immediately following the Ex N-terminal FERM domain ($^{452}$**TSG**IV**S**$^{457}$). In agreement with the fact that β-TrCP targets substrates for ubiquitylation and degradation through the recognition of a phosphodegron, Ex is phosphorylated in *Drosophila* S2 cells in the presence of ectopic Crb (full-length or the intracellular domain, Crb$^{intra}$) (*Ling et al., 2010*; *Robinson et al., 2010*; *Ribeiro et al., 2014*). However, the kinase(s) involved in Ex degradation downstream of Crb are currently unknown. In our previous report, we used affinity purification coupled with mass spectrometry (AP-MS) to identify Slmb as an Ex interacting protein (*Ribeiro et al., 2014*). Upon re-analysis of our AP-MS data, we observed that Gilgamesh (Gish), the *Drosophila* orthologue of Casein kinase 1γ, was specifically purified by an Ex truncation that fully recapitulates the Crb-mediated effect on Ex stability (Ex$^{1\text{-}468}$) (*Figure 1A*) (*Ribeiro et al., 2014*). Importantly, Gish peptides were detected in the Ex$^{1\text{-}468}$ AP-MS only upon co-expression with wild-type (wt) Crb but not with a FERM-binding motif mutant version of Crb (ΔFBM) that cannot bind Ex or promote its depletion

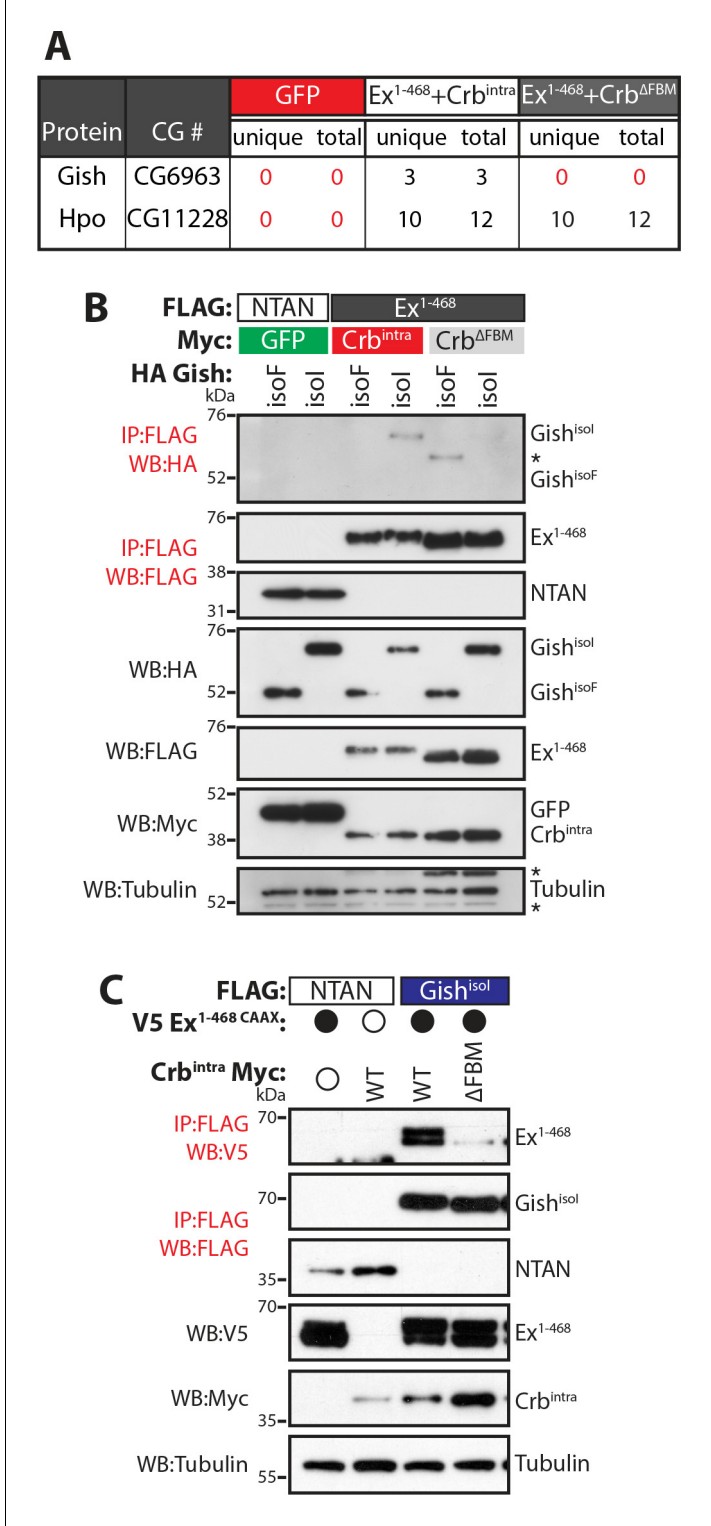

**Figure 1.** Gish, the *Drosophila* orthologue of Ckɪγ interacts with Ex in a Crb-dependent manner. (**A**) An AP-MS approach identified Gish as an Ex-interacting protein. Summary table with AP-MS results for Gish and Hpo. CG# denotes Flybase CG number, while unique and total denote the number of peptides detected in the MS analysis. (**B**) and (**C**) Crb[intra] promotes Ex:Gish binding in a FBM-dependent manner. Reciprocal co-IPs were performed with FLAG-tagged NTAN or Ex[1-468] and HA-tagged Gish (**B**), or with FLAG-tagged NTAN or Gish[isol] and V5-tagged Ex[1-468 CAAX] (**C**), in the presence of Myc-tagged GFP, Crb[intra] or Crb[ΔFBM]. The expression and presence of co-purified proteins were analysed by immunoblotting with the indicated antibodies. Asterisks denote non-specific
*Figure 1 continued on next page*

*Figure 1 continued*
bands (IgG heavy chain in IP panel and FLAG signal in Tubulin panel). Open and full circles denote absence or presence of the indicated plasmid, respectively. Tubulin was used as loading control. Note that experiments shown in (C) were performed in the presence of proteasome inhibitors.
DOI: https://doi.org/10.7554/eLife.46592.002
The following figure supplement is available for figure 1:

**Figure supplement 1.** Evolutionary conservation of CKIs and features of Gish.
DOI: https://doi.org/10.7554/eLife.46592.003

(*Figure 1A*) (*Ling et al., 2010*; *Robinson et al., 2010*; *Ribeiro et al., 2014*). In contrast, Hpo interacted with Ex regardless of Crb presence, in agreement with previous reports (*Figure 1A*) (*Yu et al., 2010*).

We validated our AP-MS data by performing co-immunoprecipitation (co-IP) analyses in S2 cell lysates. In reciprocal co-IP experiments, an interaction between Gish and Ex$^{1-468}$ was readily detected in S2 cells, specifically in the presence of wt Crb (*Figure 1B–C*). The Casein kinase 1 (CKI) protein family includes several members (*Jiang, 2017*), with high homology in their kinase domains (*Figure 1—figure supplement 1A–C*), which are thought to, at least in part, share consensus sequences and targets (*Knippschild et al., 2014*; *Schittek and Sinnberg, 2014*; *Venerando et al., 2014*). The *Drosophila gish* (CKIγ) locus produces multiple protein isoforms, one of which, Gish$^{isoF}$, lacks a conserved C-terminal palmitoylation motif (*Figure 1—figure supplement 1B,D*), which would be predicted to affect its localisation (*Davidson et al., 2005*; *Li et al., 2016*). Therefore, we used two different isoforms of Gish that either contain (Gish$^{isoI}$) or lack (Gish$^{isoF}$) this palmitoylation sequence. Interestingly, only Gish$^{isoI}$, but not Gish$^{isoF}$, was able to interact with Ex (*Figure 1B*) suggesting that CKI sub-cellular localisation may be critical to the regulation of Ex.

## CKI kinases promote Ex phosphorylation and degradation in vitro

The CKI family commonly targets substrates for proteasomal degradation (*Knippschild et al., 2014*; *Schittek and Sinnberg, 2014*), prompting us to test if Gish could promote Ex degradation. To this end, we analysed Ex electrophoretic mobility shift and protein levels (indicative of phosphorylation and degradation, respectively) in S2 cells upon expression of different CKI kinases. Co-expression of Ex$^{1-468}$ and Gish$^{isoI}$ in the absence of Crb resulted in a mild depletion of Ex$^{1-468}$ protein levels (*Figure 2A*), whereas Gish$^{isoF}$ had no effect. We reasoned that the mild effect of Gish$^{isoI}$ on Ex stability might be due to relatively low levels of Ex$^{1-468}$ reaching the membrane in the absence of Crb, which is not endogenously expressed in S2 cells. To mimic the apical membrane localisation of endogenous Ex, we generated an Ex variant containing a C-terminal CAAX sequence (*Sotillos et al., 2004*), which targets Ex to cellular membranes (Ex$^{1-468\ CAAX}$). Importantly, this variant remains responsive to Crb (*Figure 2B*), and co-immunoprecipitates with Gish in a Crb-dependent manner (*Figure 1C*). Ex$^{1-468\ CAAX}$ ran mostly as a single band in S2 cells in the presence of the Crb$^{\Delta FBM}$ mutant and, similar to wt Ex$^{1-468}$, as a higher mobility band or doublet in the presence of wt Crb (*Figure 2A–C*). An electrophoretic mobility shift and depletion was also observed when Ex$^{1-468\ CAAX}$ was co-expressed with Gish$^{isoI}$, but not with a kinase-dead version of Gish (Gish$^{KD}$) (*Figure 2B* and *Figure 2—figure supplement 1A*). Next, we tested whether two other CKI family members, CkIα and Dco (*Drosophila* CkIδ/ε orthologue), had a similar effect on Ex and whether this effect was dependent on their kinase activity. Indeed, co-expression of Ex$^{1-468\ CAAX}$ or full-length (FL) Ex with wt Gish$^{isoI}$, CkIα or Dco led to Ex mobility shift and depletion, although their kinase-deficient (KD) versions did not (*Figure 2C* and *Figure 2—figure supplement 1A–B*). Together, these results suggest that multiple CKI family kinases can promote Ex phosphorylation and depletion.

The fact that Ex and Gish interact, and that CKI kinases stimulate Ex degradation in a kinase-dependent manner led us to examine if the CKIs act in the same pathway as Crb and Slmb. To test this, we co-expressed CkIα, Dco or Gish$^{isoI}$ in S2 cells with either wt Ex$^{1-468\ CAAX}$ or a variant carrying a mutation at S453 (Ex$^{1-468\ S453A\ CAAX}$), one of the crucial residues that mediates the interaction between Ex and Slmb (*Ribeiro et al., 2014*). All three kinases readily promoted the degradation of wt Ex$^{1-468\ CAAX}$ (*Figure 2D*). However, the S453A mutant levels were largely unaffected by CKI isoforms, consistent with the notion that the CKI-mediated regulation of Ex stability is dependent on

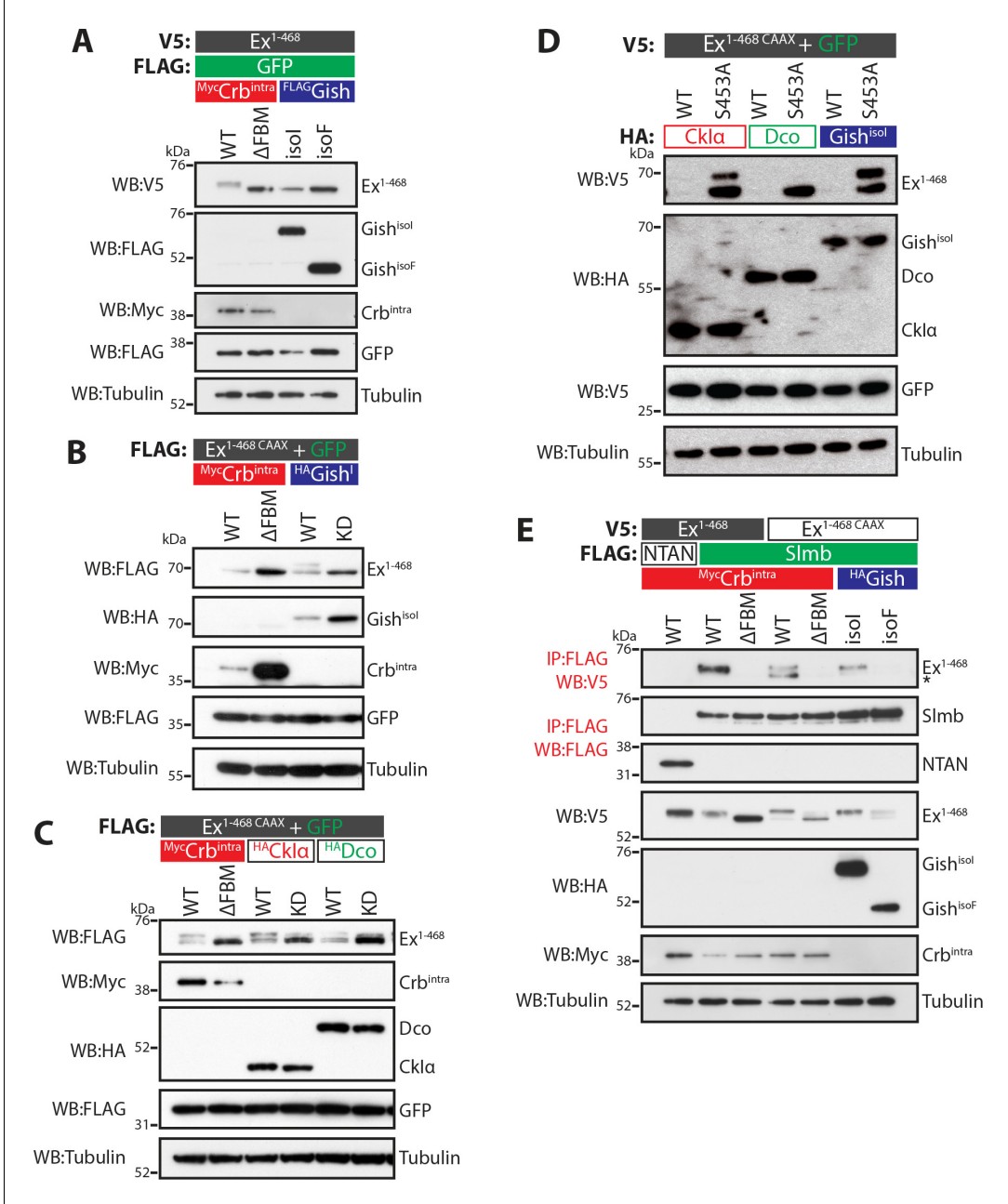

**Figure 2.** CKI kinases promote Ex phosphorylation and depletion. (**A**) and (**B**) Gish promotes Ex phosphorylation and depletion in a kinase- and sub-cellular localisation-dependent manner. (**A**) S2 cells were used to co-transfect V5-tagged Ex[1-468] with GFP and Crb[intra], Crb[ΔFBM], Gish[isol] or Gish[isoF], for 48 hr before lysis. Lysates were processed for immunoblot analysis using the indicated antibodies. Note that Gish[isol] caused a more prominent depletion of Ex than Gish[isoF], the CkIγ isoform lacking a palmitoylation sequence. (**B**) FLAG-tagged Ex[1-468 CAAX] was co-transfected with the indicated plasmids for 48 hr and lysates were processed for Western blotting analysis with the indicated antibodies. Kinase-deficient (KD) Gish[isol] was unable to promote Ex[1-468 CAAX] phosphorylation and depletion. (**C**) CKI kinases promote Ex phosphorylation and depletion in a kinase-dependent manner. S2 cells were transfected with the indicated plasmids for 48 hr before lysis. Immunoblot analysis with the indicated antibodies revealed that CkIα and Dco promote Ex phosphorylation and depletion in a kinase-dependent manner, as the kinase-dead (KD) versions did not cause a mobility shift in Ex[1-468 CAAX]. (**D**) CKI kinases promote Ex degradation via S453. S2 cells were co-transfected with V5-tagged wt or S453A Ex[1-468 CAAX] and HA-tagged CkIα, Dco or Gish[isol] for 48 hr before lysis. Lysates were immunoblotted with the indicated antibodies. Unlike its wt counterpart, Ex[1-468 S453A CAAX] was refractory to the action of CKI kinases and was not degraded in the presence of the kinases. (**E**) Gish expression promotes Ex:Slmb binding in the absence of Crb[intra]. Co-IPs were performed with FLAG-tagged NTAN or Slmb and either V5-tagged Ex[1-468] or Ex[1-468 CAAX], in the presence of Crb[intra], Crb[ΔFBM], Gish[isol] or Gish[isoF]. Expression and presence of co-immunoprecipitated proteins was assessed by immunoblotting with the indicated antibodies. Note

*Figure 2 continued on next page*

*Figure 2 continued*

that, similar to Crb[intra], Gish[isol] expression alone promoted the Ex:Slmb interaction. GFP and tubulin were used as transfection efficiency and loading controls, respectively. Asterisk denotes non-specific band (IgG heavy chain in IP panel).

DOI: https://doi.org/10.7554/eLife.46592.004

The following source data and figure supplements are available for figure 2:

**Figure supplement 1.** Molecular requirements for the effect of CKIs in the regulation of Ex protein stability.

DOI: https://doi.org/10.7554/eLife.46592.005

**Figure supplement 1—source data 1.** Source data for quantification of Expanded protein levels from Western blot analyses performed in *Drosophila* S2 cells.

DOI: https://doi.org/10.7554/eLife.46592.006

the ability of Ex to interact with Slmb via the phosphodegron surrounding S453. Interestingly, expression of CkIα or Gish[isol] resulted in the appearance of a slower-migrating Ex[1-468 S453A] band (*Figure 2D*). This suggests that, at least for CkIα and Gish, there may be alternative CKI phosphorylation sites in Ex[1-468] besides S453.

We also assessed whether CKI kinases could promote the Ex:Slmb interaction, which we have previously shown can be triggered by Crb over-expression (*Ribeiro et al., 2014*). We found that Gish[isol], but not Gish[isoF], promoted the interaction between Ex[1-468 CAAX] and Slmb in the absence of ectopic Crb (*Figure 2E*), consistent with its ability to stimulate Ex mobility shift and depletion. CkIα and Dco were also able to induce binding between Ex[1-468 CAAX] and Slmb in the absence of Crb (*Figure 2—figure supplement 1C*). Dco consistently showed a weaker effect on Ex phosphorylation than the other two kinases. Our previous findings showed that the Crb-induced Ex:Slmb interaction relies on the region surrounding the β-TrCP consensus motif (aa 450–468 in Ex) (*Ribeiro et al., 2014*). Interestingly, an Ex truncation mutant lacking this domain (Ex[1-450]) was refractory to degradation, and did not display a mobility shift following Gish[isol] expression (*Figure 2—figure supplement 1D*), suggesting that CKIs act upstream of Slmb to promote Ex phosphorylation and degradation.

β-TrCP substrates are often targeted for degradation by the sequential action of GSK3β and CKI kinases (e.g. β-catenin) (*Gammons and Bienz, 2018*). Thus, we tested if the *Drosophila* GSK3β orthologue Shaggy (Sgg) was involved in the regulation of Ex phosphorylation (*Siegfried et al., 1992*). However, we found that over-expression of Sgg did not promote Ex:Slmb binding (*Figure 2—figure supplement 1E*). Moreover, RNAi-mediated depletion of *sgg* did not abrogate Crb-induced Ex phosphorylation and degradation (*Figure 2—figure supplement 1F–G*). Together, our results indicate that the CKI kinases, but not Sgg, regulate Ex phosphorylation and stability, acting upstream of Slmb/β-TrCP.

## CKI kinases promote Ex phosphorylation and degradation in vivo

Next, we wanted to validate our observations in developing *Drosophila* tissues. However, assessing Ex protein stability in vivo is complicated by the fact that *ex* is a Yki target gene (*Hamaratoglu et al., 2006*) and, therefore, its steady state protein levels reflect not only direct post-translational effects but also transcriptional inputs via the modulation of Hpo signalling. To directly study Ex stability in the absence of any confounding effects due to Yki-mediated transcriptional feedback, we generated transgenic flies carrying an Ex stability reporter construct consisting of Ex[1-468] fused to GFP, whose expression is controlled by the *ubiquitin 63E* promoter rather than its endogenous promoter (ubi-Ex[1-468]::GFP). We also generated a mutant version that is refractory to Crb-induced degradation by mutating Ser 453, which mediates binding to Slmb (*Ribeiro et al., 2014*). As predicted, the ubi-Ex[1-468]::GFP reporters resemble endogenous Ex in that they are normally localised at the apical cortex of wing imaginal disc cells (*Figure 3A,B,D,F* and *Figure 3—figure supplement 1A–B*). Importantly, the in vivo reporters recapitulate the effect of Crb on Ex protein stability and localisation. In agreement with previous observations of Ex[FL], ubi-Ex[1-468]::GFP failed to localise to the apical surface in *crb* mutant tissue (*Figure 3A–A'* and *Figure 3—figure supplement 1A*) and accumulated at the apical cortex in *slmb* mutant clones (*Figure 3B–B'* and *Figure 3—figure supplement 1B*) (*Chen et al., 2010*; *Hafezi et al., 2012*; *Ling et al., 2010*; *Robinson et al., 2010*; *Ribeiro et al., 2014*). Loss of both *crb* and *slmb* led to accumulation of Ex[1-468]::GFP in the cytoplasm (*Figure 3C–C'*), suggesting that the Ex sensor can be degraded by

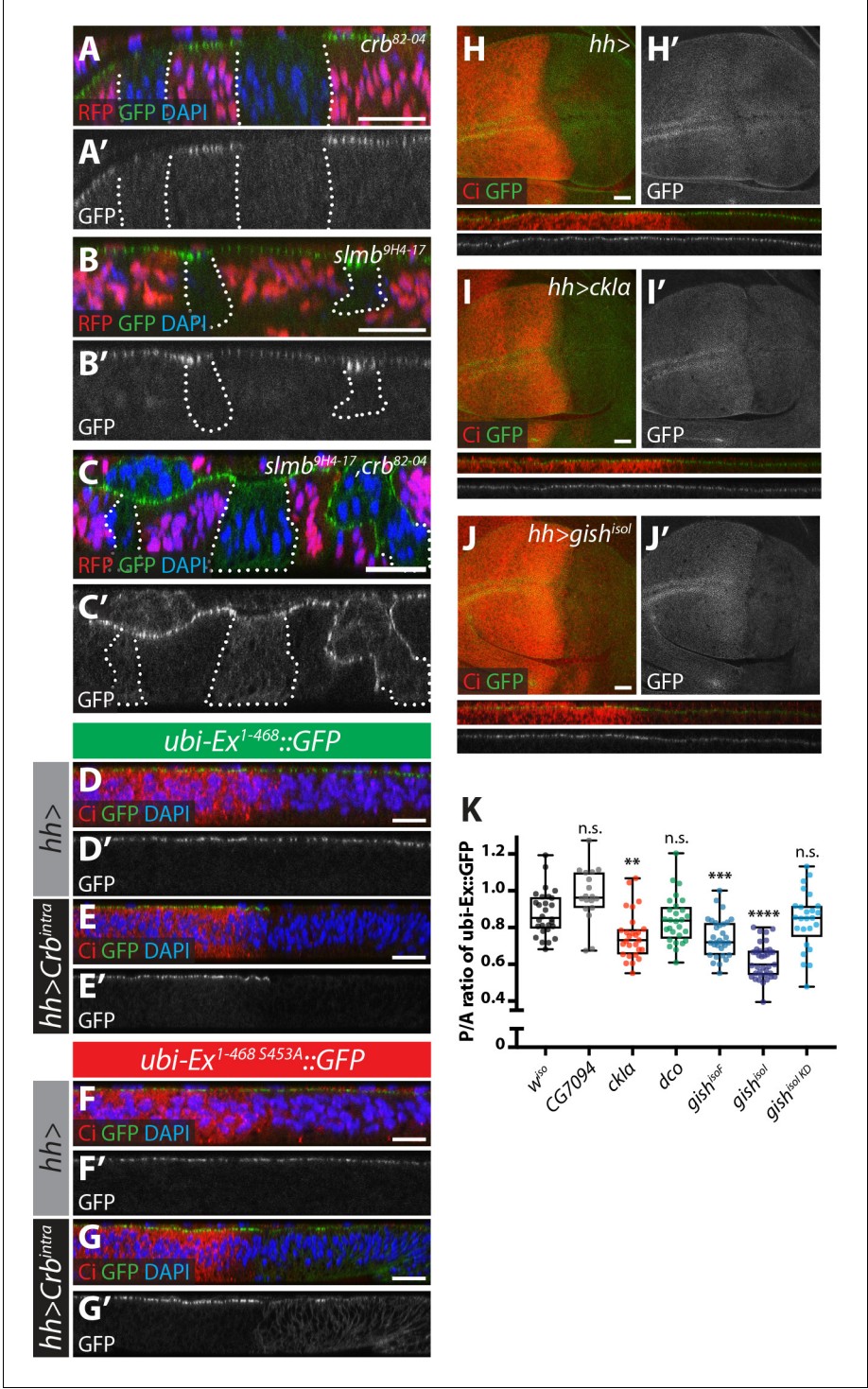

**Figure 3.** Crb-, Slmb- and CKI-mediated regulation of an in vivo Ex protein stability reporter. (**A–C**) Crb and Slmb regulate localisation and in vivo levels of the *ubi-Ex*[1-468]*::GFP* reporter. Confocal micrographs of transverse sections of wing imaginal discs from wandering third instar larvae containing clones mutant for *crb*[82-04] (**A**), *slmb*[9H4-17] (**B**), or doubly mutant for *slmb*[9H4-17] and *crb*[82-04] (**C**). Clones are marked by absence of RFP and highlighted by white dashed lines. The *ubi-Ex*[1-468]*::GFP* reporter (green in A-C, grey in A'-C') is lost from the apical cortex of *crb* clones, accumulates apically in *slmb* clones, and accumulates in the cytoplasm of *slmb*, *crb* clones. DAPI staining (blue) marks nuclei. (**D–G**) Effect of Crb[intra] on the in vivo Ex protein stability reporter. Confocal micrographs of transverse sections of wing discs from wandering third instar larvae expressing a wt (**D and E**) or a S453A mutant version of the *ubi-Ex*[1-468]*::GFP* reporter (**F and G**) (green in D-G and grey in D'-G'), in the absence

*Figure 3 continued on next page*

*Figure 3 continued*

(**D and F**) or presence (**E and G**) of *hh-Gal4*-driven *Crb^intra* over-expression. Ci immunostaining (red) indicates the anterior compartment, where *hh* is not expressed. DAPI nuclear staining is shown in blue. In the absence of Crb^intra, both versions of the reporter localise at the apical surface. Expression of Crb^intra causes depletion of the wt reporter, while it promotes mislocalisation of the S453A variant, in agreement with previously published data (*Ribeiro et al., 2014*). (**H–K**) Over-expression of CKI kinases promotes depletion of the Ex stability reporter. XY and transverse sections of third instar wing imaginal discs expressing *ubi-Ex^1-468::GFP* (green in H-J, grey in H'-J') and either no transgene (**H**), *UAS-ckIα* (**I**) or *UAS-gish^isol* (**J**) under the control of *hh-Gal4*. Ci immunostaining (anterior compartment, lacking *hh* expression) is shown in red. Expression of CkIα or Gish reduces the levels of the Ex in vivo stability reporter in vivo. (**K**) Quantification of the ratio between the levels of the *ubi-Ex^1-468::GFP* reporter in the posterior versus the anterior compartment in wing discs of the indicated genotypes (transgene expression driven by *hh-Gal4* as in H-J). Data are shown in box (median and $25^{th}$-$75^{th}$ percentile) and whiskers (minimum to maximum) plots with all data points represented. n > 18 for all genotypes. Significance was assessed using a one-way ANOVA comparing all genotypes to $w^{iso}$, with Dunnett's multiple comparisons test. **, p<0.01; ***, p<0.001, ****, p<0.0001. n.s. non-significant. In XY sections, ventral is up, whereas apical is up in transverse sections. Scale bars, 10 μm in A-C and 20 μm in D-J.

DOI: https://doi.org/10.7554/eLife.46592.007

The following source data and figure supplements are available for figure 3:

**Source data 1.** Source data for quantification of relative protein levels of Ex::GFP in vivo reporter.
DOI: https://doi.org/10.7554/eLife.46592.010

**Figure supplement 1.** Characterisation of the Ex stability reporter and in vivo regulation of Ex stability and tissue growth.
DOI: https://doi.org/10.7554/eLife.46592.008

**Figure supplement 1—source data 1.** Source data for quantification of adult wing sizes and relative protein levels of Crb and Ex S453A::GFP reporter.
DOI: https://doi.org/10.7554/eLife.46592.009

SCF^Slmb/β-TrCP in the cytoplasm, as had been suggested for endogenous Ex (*Zhang et al., 2015*). When a Crb^intra transgene was expressed in the posterior compartment using the *hedgehog-Gal4* driver (*hh-Gal4*), ubi-Ex^1-468::GFP was lost from the apical surface and degraded (*Figure 3D–E* and *Figure 3—figure supplement 1C*). In contrast, the S453A mutant reporter accumulated upon Crb^intra expression, presumably due to the fact that it cannot be recognised by Slmb and degraded (*Figure 3F–G* and *Figure 3—figure supplement 1D*). This re-localisation of Ex^1-468 S453A mirrored the localisation of over-expressed Crb^intra, which localised to the apical cortex as well as basal membranes (*Figure 3—figure supplement 1E*). It is therefore possible that Ex^1-468 S453A is mislocalised to the basolateral membrane by ectopic Crb, where it cannot be degraded. A further advantage of using this N-terminal fragment as a reporter is that our constructs lack the Yki-binding PPxY motifs, and therefore, do not induce cell death and reduced tissue growth, as does full length Ex (*Hamaratoglu et al., 2006*; *Badouel et al., 2009*; *Oh et al., 2009*). Indeed, neither reporter affected wing size when expressed at low levels under the control of the *ubi* promoter (*Figure 3—figure supplement 1F–I*). In contrast, expression of full-length Ex under the control of *en-Gal4* driver induced a reduction in tissue size (*Figure 3—figure supplement 1J,L,P*). This phenotype was partially rescued by co-expression of Crb^intra, which we expect would promote increased turnover and degradation of Ex (*Figure 3—figure supplement 1M,P*). Interestingly, expression of the S453A mutant version of Ex resulted in a more severe undergrowth than wt Ex, and this was refractory to Crb^intra co-expression, suggesting that Crb fails to regulate the levels of Ex when the residue that is recognised by Slmb/β-TrCP is mutated (*Figure 3—figure supplement 1N–P*).

We then analysed whether CKI over-expression could modulate levels of the Ex reporter. Indeed, when compared to control wing discs, expression of CkIα or Gish using *hh-Gal4* resulted in a significant decrease in the levels of the reporter in the posterior compartment (*Figure 3H–3K*). This is consistent with our S2 cell results, and suggests that the CKI family kinases regulate Ex stability in vivo. However, this effect on Ex levels was not seen upon over-expression of either Dco or the poorly characterised CKI family member CG7094 (*Figure 3K*), indicating that not all CKIs modulate Ex stability in vivo. Importantly, the effect of CKIs on Ex is not due to changes in Crb levels (*Figure 3—figure supplement 1Q*), and is dependent on kinase activity, since a Gish^isol kinase-deficient construct could not alter reporter levels (*Figure 3K*). In contrast, over-expression of CKIs did not alter the

levels of the S453A mutant reporter (*Figure 3—figure supplement 1R*), providing further evidence that CKIs regulate Ex in a Slmb/β-TrCP-dependent manner.

## CKI loss-of-function promotes stabilisation of Ex in vitro

Next, we sought to analyse loss-of-function phenotypes of the CKI kinases. Upon treatment of S2 cells with *gish* RNAi, Crb$^{intra}$-mediated depletion of both Ex$^{1-468}$ and Ex$^{FL}$ was strongly reduced (*Figure 4A–B* and *Figure 4—figure supplement 1A*). We also attempted to deplete the other CKI kinases by RNAi. However, we were unable to generate a dsRNA that specifically targeted *dco* (*Figure 4—figure supplement 1B*). To overcome this, we used RNAi sequences that targeted only *cklα* ("*cklα*") or all CKIs (*cklα*, *dco* and *gish*, termed "*ckl$^{pan}$*", previously used in *Liu et al. (2002)* (*Figure 4—figure supplement 1B*). In line with our previous data, knockdown of *cklα* or *gish* alone resulted in robust stabilisation of Ex$^{1-468}$ in the presence of Crb$^{intra}$ (*Figure 4C*). It is noteworthy that depleting all CKIs resulted in a more prominent stabilisation of Ex$^{1-468}$, suggesting that Ckiα and Gish work together to regulate Ex stability (*Figure 4C*).

As the CKI kinases appear to act downstream of Crb to regulate Ex stability, we also assessed if their depletion could affect the ability of Crb to promote Ex:Slmb binding and, consequently, Slmb-mediated Ex ubiquitylation and degradation. Indeed, the interaction between Ex and Slmb was abolished when *gish* or when all CKIs were depleted by RNAi (*Figure 4D*). When we monitored Crb-induced Ex ubiquitylation, we observed that this was dramatically reduced when *gish* or all CKIs were depleted by RNAi (*Figure 4E–F*). Indeed, knocking-down CKIs had a similar effect to depletion of *slmb*, part of the Ex E3 ligase complex activated downstream of Crb (*Figure 4E*). These data suggest that CKI kinases act downstream of Crb to promote the interaction of Ex with Slmb, thereby stimulating Ex ubiquitylation and degradation.

Phosphorylation of Crb by aPKC has been proposed to influence its function and, therefore might contribute to the regulation of Ex (*Ribeiro et al., 2014*; *Sotillos et al., 2004*). To test this, we compared the effect of wt Crb to that of a variant of Crb containing point mutations in the putative aPKC phosphorylation sites (Crb$^{4A\ mut}$). Similar to wt Crb, Crb$^{4A\ mut}$ efficiently promoted Ex degradation (*Figure 4—figure supplement 1C*), suggesting that aPKC phosphorylation is not required for the regulation of Ex downstream of Crb. A previous report has shown that, in addition to the N-terminal Slmb/β-TrCP degron, there is a C-terminal degron in Ex, which is inhibited by Wts-mediated phosphorylation (*Zhang et al., 2015*). To investigate whether Wts plays a role in Crb-mediated Ex degradation, we co-expressed Ex$^{FL}$ with either Crb$^{FL}$ or Crb$^{intra}$, in the presence or absence of ectopic Wts (*Figure 4—figure supplement 1D*). Though our data confirm that Wts can stabilise Ex$^{FL}$, we observed that the presence of Wts did not prevent the degradation of Ex mediated by the expression of Crb, indicating that the N-terminal degron supersedes the C-terminal one when Crb triggers Ex degradation (*Figure 4—figure supplement 1D*).

## Clonal analysis of CKI mutants

As CKI loss-of-function in S2 cells resulted in Ex stabilisation, we next aimed to validate these observations in vivo. When we assessed the levels of the ubi-Ex$^{1-468}$::GFP reporter in third instar wing imaginal discs carrying clones mutant for existing *gish* (*gish$^{KG03891}$*) or *dco* (*dco$^{le88}$*) alleles, we observed no differences from control cells, suggesting that these kinases are not essential for maintaining Ex levels at steady state in vivo (*Figure 5A* and *Figure 5—figure supplement 1A*). We observed occasional *gish$^{KG03891}$* adult escapers, leading us to hypothesise that this allele is hypomorphic. Therefore, we generated a new *gish* mutant (bearing a premature termination codon at the N-terminal part of the kinase domain), as well as a mutant for the poorly characterised CKI family member CG7094, which is also expressed in imaginal discs (*Brown et al., 2014*) using CRISPR/Cas9 gene editing (*Figure 5—figure supplement 1B–C*). However, when we generated FRT clones for these new alleles, neither caused a change in ubi-Ex$^{1-468}$::GFP levels compared to wt tissue (*Figure 5B* and *Figure 5—figure supplement 1D*). In contrast, we observed that the ubi-Ex$^{1-468}$::GFP reporter was elevated in *cklα$^{8B12}$* clones (*Figure 5C*). We noticed that apical levels of Crb were also increased in *cklα$^{8B12}$* clones (*Figure 5—figure supplement 1E*). This was not due to a general increase in apical domain size, since levels of another apical protein, aPKC, did not increase in *cklα$^{8B12}$* clones (*Figure 5—figure supplement 1F–G*).

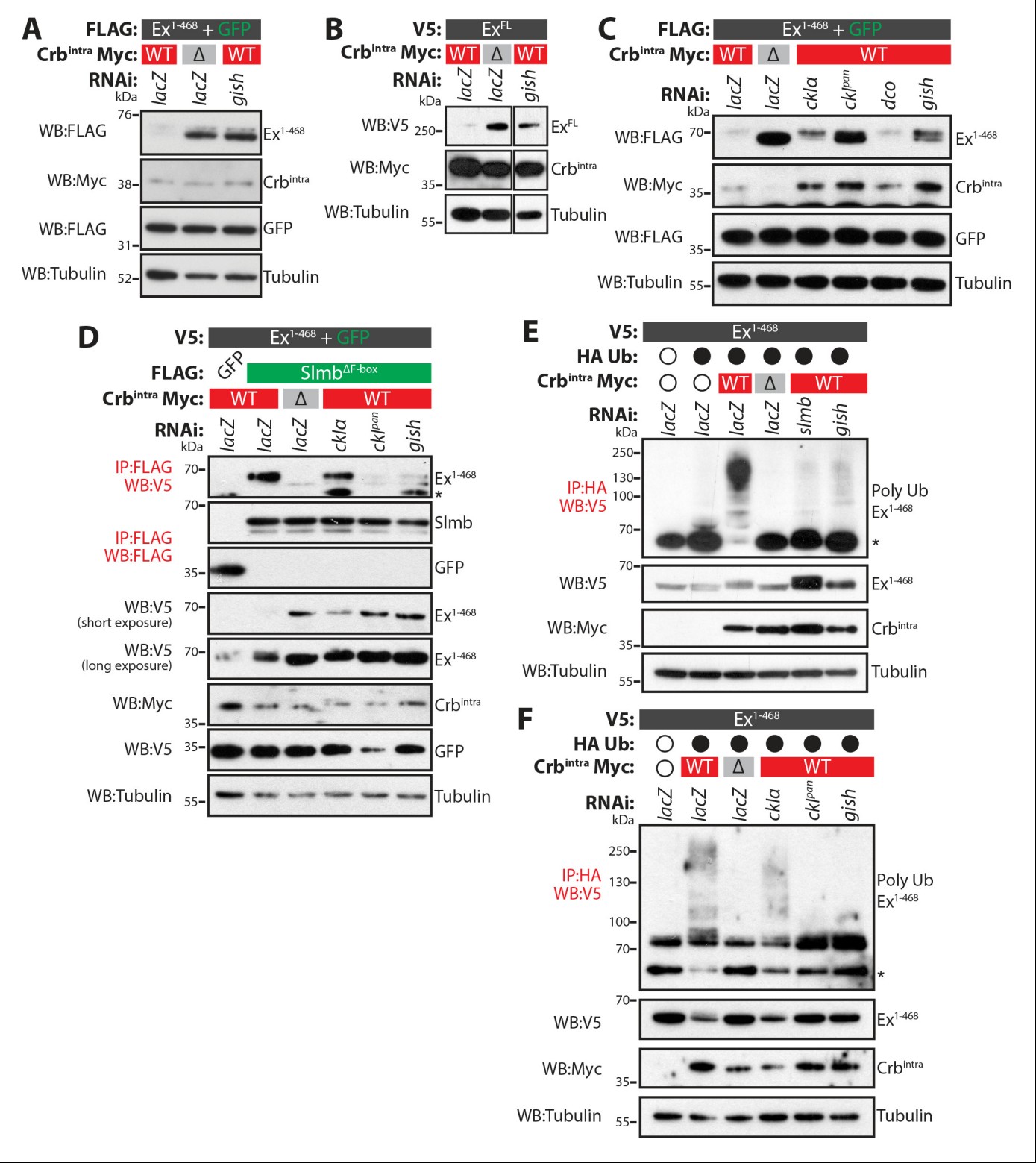

**Figure 4.** Cklα and Gish are required for Crb-induced Ex degradation. (**A**) and (**B**) RNAi-mediated depletion of *gish* abrogates Crb-induced Ex degradation. FLAG-tagged Ex[1-468] (**A**) or V5-tagged Ex[FL] (**B**) were co-expressed with Crb[intra] or Crb[ΔFBM], in the presence of dsRNA targeting *lacZ* (control) or *gish*. Lysates were processed for immunoblot analysis using the indicated antibodies. In both cases, *gish* depletion blocked Ex degradation induced by Crb[intra] expression. (**C**) CKI kinase knockdown blocks Ex degradation induced by expression of Crb[intra]. S2 cells treated with dsRNA targeting *lacZ* or CKI kinases were co-transfected with FLAG-tagged Ex[1-468] and GFP, Crb[intra] or Crb[ΔFBM]. Immunoblot analysis of lysates using the

*Figure 4 continued on next page*

*Figure 4 continued*

indicated antibodies revealed that depleting *cklα* and *gish* alone or all CKIs (*ckl^pan*) dramatically impaired the ability of Crb^intra to promote Ex^1-468 degradation. (**D**) Crb requires CKI kinase function to promote Ex:Slmb binding. Co-IPs were performed between FLAG-tagged GFP or Slmb^ΔF-box and V5-tagged Ex^1-468 in the presence of Crb^intra or Crb^ΔFBM and depletion of *lacZ* or CKI kinases. Lysates were analysed by immunoblot using the indicated antibodies for detection of protein expression and co-purification. Note that depletion of all CKIs or *gish* prevented the Ex:Slmb interaction induced by Crb^intra. (**E**) and (**F**) CKI kinases are required for Crb^intra-induced Ex ubiquitylation. S2 cells were treated with the indicated dsRNAs for 24 hr before transfection with the indicated constructs. Following lysis under denaturing conditions, ubiquitylated proteins were isolated using anti-HA antibodies. The presence of Ex and Crb^intra was assessed with the indicated antibodies. Knockdown of *gish* (**E and F**) or of all CKIs with *ckl^pan* RNAi (**F**) significantly reduced Ex ubiquitylation, similar to depletion of *slmb*. GFP was used as transfection efficiency control. Tubulin was used as loading control. Open and full circles denote absence or presence of the indicated plasmid, respectively. Asterisks denote non-specific bands (IgG heavy chain in IP panels).
DOI: https://doi.org/10.7554/eLife.46592.011

The following figure supplement is available for figure 4:

**Figure supplement 1.** Role of CKI kinases and alternative kinases in the regulation of Ex stability.
DOI: https://doi.org/10.7554/eLife.46592.012

---

It is possible that the increase in Ex levels in *cklα^8B12* clones is the result of an increase in Crb within the clones, causing enhanced recruitment of Ex to the apical cortex, rather than an effect on Ex stability. Alternatively, increased Ex levels as a result of its stabilisation in *cklα^8B12* clones might in turn lead to increased membrane retention or stabilisation of Crb. If this were the case, increasing Ex dosage should increase Crb levels. Indeed, over-expression of UAS-Ex^1-468 led to a robust increase in endogenous Crb at the apical membrane (*Figure 5D*). We had previously reported that clones mutant for the hypomorphic allele *slmb^1* showed elevated levels of Ex, but not Crb (*Ribeiro et al., 2014*). However, upon careful examination, we were able to detect increased Crb levels in *slmb^1* clones, albeit mainly when these clones were large (*Figure 5E*). Furthermore, mutant clones for a null allele, *slmb^9H4-17*, displayed a clear increase in Crb levels (*Figure 5F*). Thus, both *slmb* and *cklα* mutant clones display elevated levels of Ex and Crb.

Since our S2 cell data suggested that several members of the CKI family may regulate Ex redundantly (*Figure 4C*), we generated triple mutant clones lacking *CG7094, gish and dco*, but these did not have altered levels of the Ex^1-468::GFP reporter (*Figure 5—figure supplement 1H*). Triple mutant clones for *cklα, gish* and *dco* had elevated Ex and Crb levels (*Figure 5—figure supplement 1I*), but these were comparable to *cklα* clones alone (*Figure 5C* and *Figure 5—figure supplement 1E–F*). Thus, in vivo, it does not appear that CKI family members function redundantly to control Ex levels, at least in wing discs under normal developmental conditions. To test whether the regulation of Ex levels by Slmb and CkIα occurs in tissues other than the wing disc, we generated mutant clones in the eye, leg and haltere imaginal discs (*Figure 5—figure supplement 2*). In the eye disc, loss of either *slmb* (*Figure 5—figure supplement 2A–A'*) or *cklα* (*Figure 5—figure supplement 2D–D'*) led to a strong clone extrusion phenotype and a loss of ommatidial differentiation posterior to the morphogenetic furrow. We did not observe a strong elevation of the Ex::GFP reporter in these mutant clones. However, as the Ex::GFP signal was most prominent in the apical domain of the ommatidial units, it is difficult to conclude if Slmb and CkIα regulate Ex stability in the absence of differentiated ommatidia in the mutant tissue. In contrast, Ex::GFP levels were elevated in both *slmb* (*Figure 5—figure supplement 2B–C'*) or *cklα* (*Figure 6E–6F'*) mutant clones in the leg and haltere discs. The requirement for Slmb and CkIα to regulate Ex levels therefore varies according to the tissue.

In S2 cells, depletion of *gish* or *cklα* resulted in Ex stabilisation in the presence of Crb (*Figure 4A–C*). We hypothesised that over-expressing Crb in vivo would accelerate Ex turnover, thereby providing a sensitised background to examine the effects of CKI depletion. To circumvent the pleiotropic effects of long-term knockdown in imaginal disc epithelial cells, we expressed *cklα^RNAi* or *gish^RNAi* in the absence or presence of Crb^intra using *hh-Gal4*, and controlled the expression temporally (for 24 hr or 48 hr) using *tub-Gal80^ts* (*Figure 6*). Quantifications for these data are presented in *Figure 6—figure supplement 1B*. Expression of Crb^intra caused depletion of all apical ubi-Ex^1-468::GFP in the posterior compartment within 24 hr (*Figure 6—figure supplement 1A*). Expression of either *cklα^RNAi* (*Figure 6A–B*) or *gish^RNAi* (*Figure 6E–F*) alone had little effect on ubi-Ex^1-468::GFP expression in the posterior compartment (*Figure 6—figure supplement 1B*). However, co-expression of *cklα^RNAi* (*Figure 6C–D*), but not *gish^RNAi* (*Figure 6G–H*), with Crb^intra resulted in a

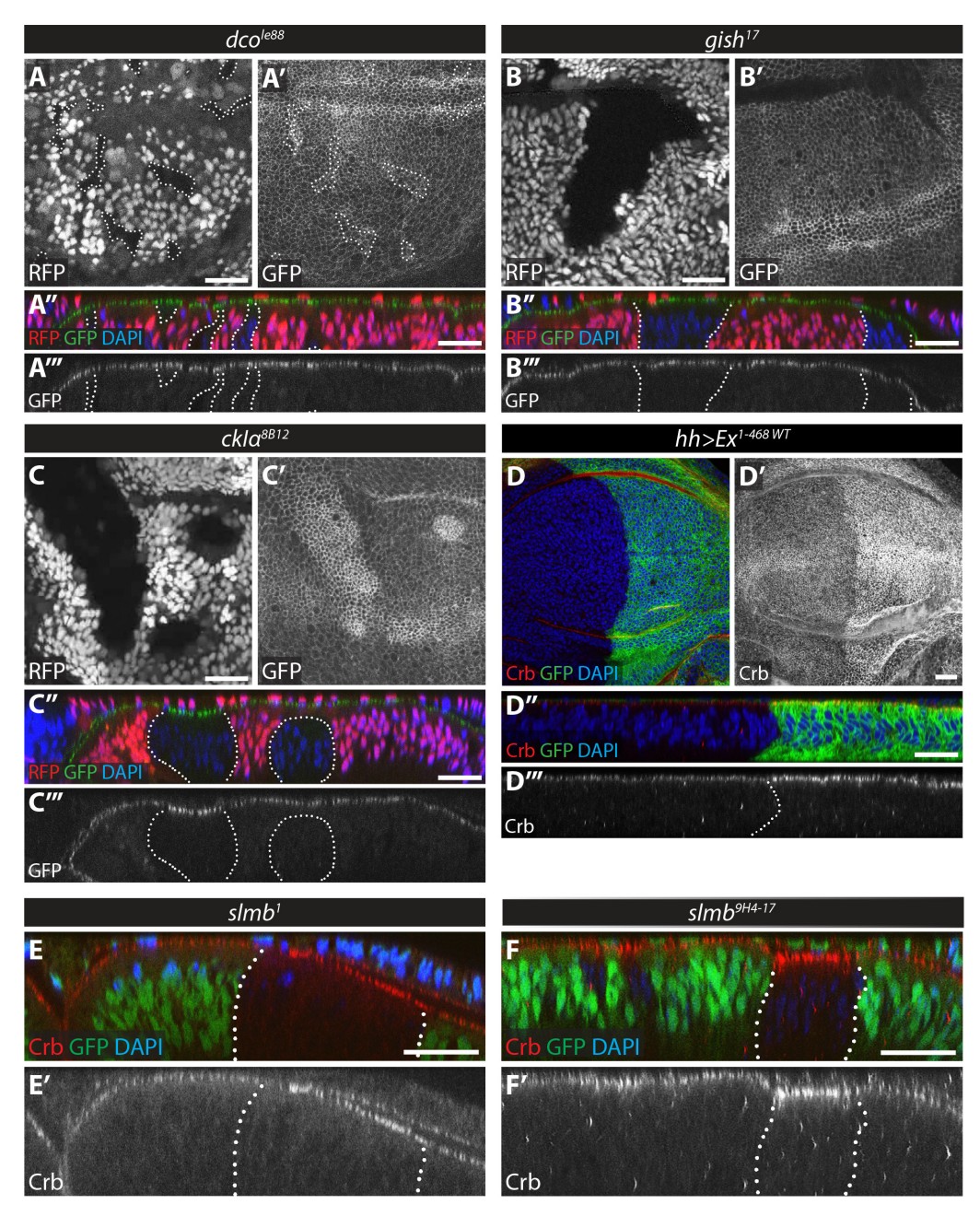

**Figure 5.** Loss of function of *cklα*, but not *gish* or *dco*, modulates levels of an Ex protein stability reporter in vivo. (A) *dco* mutant clones do not affect the Ex in vivo stability reporter. XY (A, A') and transverse sections (A'', A''') of *ubi-Ex^1-468^::GFP*-expressing third instar wing imaginal discs containing *dco^le88^* mutant clones (marked by absence of RFP and highlighted by white dashed lines), showing direct fluorescence from GFP (green in A'' and grey in A' and A''') or RFP (red in A'' and grey in A), and DAPI staining (blue). (B) *gish* mutant clones do not affect Ex^1-468^::GFP levels. XY (B, B') and transverse sections (B'', B''') of a third instar wing imaginal disc expressing *ubi-Ex^1-468^::GFP* (green in B'' and grey in B' and B''') and carrying *gish^17^* mutant clones (marked by absence of RFP and highlighted by white dashed lines) stained with DAPI (blue). (C) *cklα* loss-of-function induces higher levels of Ex^1-468^::GFP. XY (C, C') and transverse sections (C'', C''') of third instar wing imaginal discs expressing *ubi-Ex^1-468^::GFP* and carrying *cklα^8B12^* mutant clones (marked by absence of RFP and highlighted by white dashed lines). GFP reporter is shown in green (C'') or grey (C', C'''). RFP fluorescence is shown in red (C'') or grey (C) and the nuclear marker DAPI is shown in blue. (D–F) Over-expression of Ex^1-468^ or loss of *slmb* function induce higher apical levels of endogenous Crb. Shown are XY (D, D') and transverse sections (D'', D''', E, E', F and F') of third instar wing imaginal discs expressing *UAS-Ex^1-468^* under the control of *hh-Gal4* (D), or carrying loss-of-function clones for the β-TrCP alleles *slmb^1^* (E) or *slmb^9H4-17^* (F). GFP marks *hh-Gal4*-expressing domain in D, while absence of GFP expression marks *slmb* mutant clones in E and F (both highlighted by white dashed lines). Crb

*Figure 5 continued on next page*

*Figure 5 continued*

staining is shown in red (**D–F and D''**) or grey (**D'–F' and D'''**) and the nuclear marker DAPI is shown in blue. Dorsal and apical are up in XY and transverse sections, respectively. Scale bars, 20 μm.

DOI: https://doi.org/10.7554/eLife.46592.013

The following source data and figure supplements are available for figure 5:

**Figure supplement 1.** In vivo clonal analysis of the role of CKI kinases in the regulation of the Ex stability reporter.

DOI: https://doi.org/10.7554/eLife.46592.014

**Figure supplement 1—source data 1.** Source data for quantification of aPKC protein levels in wing imaginal discs carrying clones mutant forckIα.

DOI: https://doi.org/10.7554/eLife.46592.016

**Figure supplement 2.** Role of Slmb and CkIα in regulating an in vivo Ex protein stability reporter in eye, leg and haltere discs.

DOI: https://doi.org/10.7554/eLife.46592.015

significant rescue of levels and localisation of the ubi-Ex$^{1\text{-}468}$::GFP reporter compared to Crb$^{intra}$ alone (*Figure 6—figure supplement 1A–B*). Thus, *cklα* knockdown, but not *gish* knockdown, is able to inhibit Crb-mediated Ex degradation in vivo. *dco$^{RNAi}$* had no effect on Ex::GFP reporter levels in the absence (*Figure 6—figure supplement 1C*), or presence of Crb$^{intra}$ (*Figure 6—figure supplement 1D*). We confirmed these findings using the MARCM technique, and observed that Crb$^{intra}$-expressing, GFP-marked MARCM clones caused complete loss of apical ubi-Ex$^{1\text{-}468}$::mScarlet (*Figure 6—figure supplement 1E,G*). *gish* MARCM clones expressing Crb$^{intra}$ resembled Crb$^{intra}$ MARCM clones alone (loss of the apical Ex reporter) (*Figure 6—figure supplement 1F*). In contrast, removing *cklα* resulted in a partial rescue of Crb-induced Ex degradation, with a proportion of mutant cells retaining ubi-Ex$^{1\text{-}468}$::mScarlet apically (*Figure 6—figure supplement 1H*). Together, these results suggest that, in vivo, CkIα regulates Ex stability by promoting Crb-induced turnover.

An increase in Ex stability at the apical plasma membrane would be expected to lead to decreased Yki activity upon *cklα* loss (*Fulford et al., 2018*). In fact, *slmb* depletion in the posterior wing disc leads to downregulation of the Yki transcriptional reporter *ex-lacZ* and increased Yki nuclear exclusion (*Zhang et al., 2015*). Levels of both *ex-lacZ* and *diap1-GFP3.5*, a Yki-responsive fragment of the *diap1* promoter (*Zhang et al., 2008*) were markedly decreased in *cklα* mutant wing disc clones (*Figure 7A,C*), while *ex-lacZ* was not affected in *gish* mutant tissue (*Figure 7B*). Furthermore, we used a Yki-GFP knock-in line (*Fletcher et al., 2018*) to show that Yki is excluded from the nucleus in *cklα* mutant cells compared with neighbouring wild type tissue (*Figure 7D*). Finally, overexpression of CkIα, but not Gish, in the posterior compartment of the wing disc increased Yki activity as measured by *ex-lacZ* levels (*Figure 7E–H*). Thus, loss of *cklα* leads to Ex stabilisation correlated with decreased Yki activity, while its overexpression has the opposite effect, consistent with a role for CkIα in promoting Yki activity via the control of Ex stability.

## Discussion

Ex was one of the first identified upstream regulators of the Hippo pathway (*Hamaratoglu et al., 2006*) and functions as a growth suppressor in *Drosophila* (*Fulford et al., 2018*). In agreement with its key role in growth control, it is increasingly evident that Ex is tightly regulated, both at the transcriptional level via Yki itself (*Hamaratoglu et al., 2006*) and through its subcellular localisation and stability (*Chen et al., 2010*; *Ling et al., 2010*; *Robinson et al., 2010*; *Su et al., 2017*; *Ribeiro et al., 2014*; *Zhang et al., 2015*; *Ma et al., 2017*). Crb recruits Ex to its site of activity at the apical plasma membrane, but also limits its apical levels by triggering Ex turnover via Slmb/β-TrCP, ensuring the fine-tuning of Yki activity (*Chen et al., 2010*; *Ling et al., 2010*; *Robinson et al., 2010*; *Ribeiro et al., 2014*; *Zhang et al., 2015*). Slmb can also promote the degradation of Ex in the cytoplasm in a Crb-independent manner (*Zhang et al., 2015*, *Figure 3C and C'*). Another ubiquitin ligase, Plenty of SH3s (POSH) has also been implicated in Ex degradation in parallel to Slmb (*Ma et al., 2018*). Since our data indicate that Ex levels are not sensitive to Slmb in the eye imaginal disc (*Figure 5—figure supplement 2*), it is possible POSH is the dominant Ex regulator in this tissue. Interestingly, recent work indicates that the Fat atypical cadherin, a major upstream branch of Hippo signalling (*Bennett and Harvey, 2006*; *Cho et al., 2006*; *Silva et al., 2006*; *Willecke et al., 2006*) also influences Ex stability by apically recruiting the adaptor Dlish/Vamana (*Misra and Irvine, 2016*; *Wang et al., 2019*; *Zhang et al., 2016*). Dlish binds the Ex C-terminus via its SH3 domain and

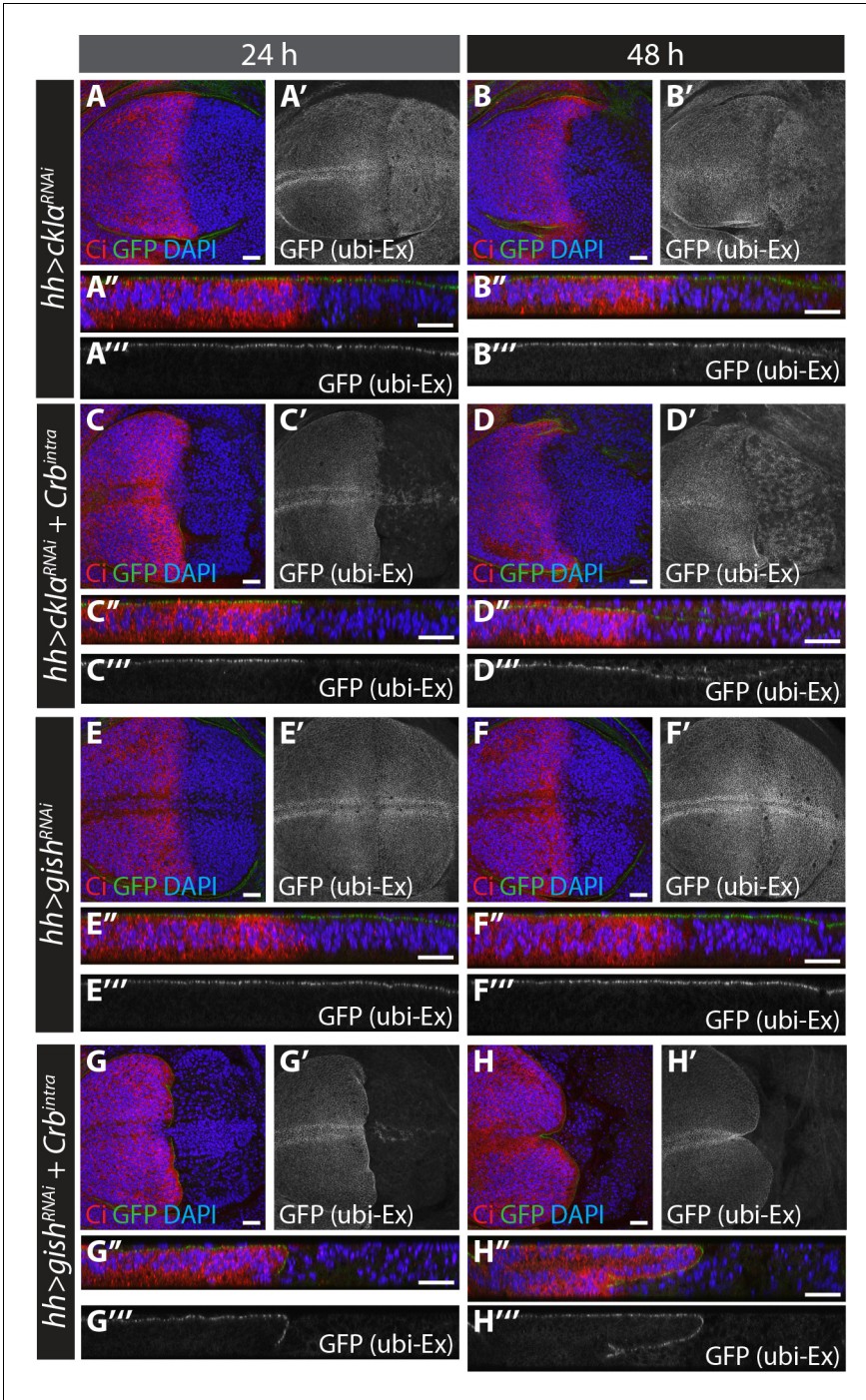

**Figure 6.** RNAi-mediated depletion of *cklα*, but not *gish*, suppresses Crb[intra]-induced degradation of an in vivo Ex protein stability reporter. (**A**) and (**B**) *cklα* knockdown has a minimal effect on *Ex[1-468]::GFP* levels.(**C**) and (**D**) *cklα* RNAi blocks Crb[intra]-mediated depletion of *Ex[1-468]::GFP*. (**E–H**) *gish* RNAi-mediated depletion does not significantly affect in vivo Ex[1-468]::GFP levels in the absence or presence of Crb[intra]. XY and transverse sections of third instar wing imaginal discs containing *ubi-Ex[1-468]::GFP*, in which *hh-Gal4* was used to drive expression of *UAS-cklα[RNAi]* alone (**A and B**), *UAS-cklα[RNAi]* and *UAS-Crb[intra]* (**C and D**), *UAS-gish[RNAi]* alone (**E and F**) or *UAS-gish[RNAi]* and *UAS-Crb[intra]* (**G and H**). Temporal control of Gal4 activity was achieved with a *tub-Gal80[ts]* transgene, raising the larvae at 25°C and shifting them to 29°C for the indicated times. *ubi-Ex[1-468]::GFP* is shown in green (**A–H** and **A''–H''**) or grey (**A'–H'** and **A'''–H'''**). Ci immunostaining (red) indicates anterior compartment where transgenes are not expressed. DAPI (blue) stains nuclei. Ventral and apical are up in XY and transverse sections, respectively. Scale bars, 20 μm.

*Figure 6 continued on next page*

*Figure 6 continued*

DOI: https://doi.org/10.7554/eLife.46592.017

The following source data and figure supplements are available for figure 6:

**Figure supplement 1.** Analysis of the effect of RNAi-mediated depletion and clonal mutation of CKI kinases in the regulation of Crb^intra^-induced Ex degradation.

DOI: https://doi.org/10.7554/eLife.46592.018

**Figure supplement 1—source data 1.** Source data for quantification of relative levels of the Ex::GFPin vivore-porter in wing imaginal discs expressing RNAi targeting CKI kinases alone or in combination with expression of Crbintra.

DOI: https://doi.org/10.7554/eLife.46592.019

promotes the Slmb:Ex association, thereby increasing Ex turnover (*Wang et al., 2019*). Thus, Ex functions as a hub in Hippo signalling that receives input from both the Crb and the Fat branches of upstream signalling.

However, recognition by Slmb/β-TrCP requires substrate phosphorylation, which had remained an unexplored aspect of Ex regulation. Here, we provide evidence that the CKI family of kinases and, in particular, CkIα and Gish, act downstream of Crb to promote phosphorylation of Ex, in turn allowing binding to Slmb/β-TrCP and Ex ubiquitylation and degradation. In conditions whereby Crb-mediated turnover of Ex is active, such as through Crb^intra^ expression, loss of *cklα* (and *gish* in cell culture) inhibited Ex degradation (*Figure 4A–C* and *Figure 6C–D*). In the absence of CKI function, Crb is unable to trigger Ex:Slmb binding and thus can no longer induce Ex degradation (*Figure 4D–F*). In agreement with the requirement for Slmb/β-TrCP function, Ex degradation is dependent on phosphorylation and the presence of a β-TrCP consensus site ($^{452}$**TSG**IVS$^{457}$), which is phosphory-lated to allow the interaction to occur (*Ribeiro et al., 2014*). Our data indicate that CKIs mediate Ex phosphorylation downstream of Crb, as the effect of CKIs on Ex stability is dependent on their kinase activity (*Figure 2B–C*, *Figure 2—figure supplement 1A–B* and *Figure 3K*) and mutation of a conserved residue in the Slmb/β-TrCP consensus (S453A) rendered Ex insensitive to degradation induced by Crb, CKIs or Slmb (*Ribeiro et al., 2014*) (*Figures 3G* and *2D*). Interestingly, CkIα and Gish were still able to induce a mobility shift in Ex when the S453 residue was mutated (*Figure 2D*), suggesting that other residues besides S453 may be targets of CKI-mediated phosphorylation. An Ex truncation lacking residues 450–468 (Ex$^{1-450}$) fails to undergo a mobility shift in the presence of Crb^intra^ or Gish (*Figure 2—figure supplement 1D*). This may indicate that the additional residues targeted by CKI are in this region, or simply that CKI needs these residues to dock onto Ex.

It remains to be determined whether phosphorylation at additional sites besides S453 is required to promote Ex:Slmb binding, and whether CKIs are part of a phospho-priming mechanism that involves additional kinases, as commonly seen for other CKI targets, such as β-catenin downstream of Wnt signalling (*Liu et al., 2002*; *Amit et al., 2002*; *Jiang and Struhl, 1998*; *Stamos and Weis, 2013*; *Winston et al., 1999*). In the case of β-catenin, CK1α acts as a priming kinase for GSK3β/Sgg; however, our data suggest that Sgg does not affect Ex stability (*Figure 2—figure supplement 1E–G*). Wts/LATS is another candidate, since mammalian LATS1/2 act as priming kinases for CK1δ/ε to target YAP for degradation by SCF$^{β-TrCP}$(*Zhao et al., 2010*). However, Wts has been suggested to stabilise Ex through phosphorylation at S1116, rather than promote its degradation (*Zhang et al., 2015*). Furthermore, Crb-dependent Ex degradation appears to override Wts-mediated stabilisation (*Zhang et al., 2015* and *Figure 4—figure supplement 1D*). The potential involvement of other kin-ases in Ex/Slmb binding therefore remains an open question.

An unexpected aspect of our results is the fact that, in cultured S2 cells, Gish (CKIγ) depletion has a strong effect on Ex stability (*Figure 4A–C*), while CkIα appears to be the dominant player in the wing imaginal disc, both in the presence (*Figure 6C–D*) and absence (*Figure 5C*) of Crb^intra^ expres-sion. This is surprising, since Crb-induced Ex phosphorylation and ubiquitylation presumably occurs at the apical plasma membrane, where Gish is known to be localised (*Morin et al., 2001*). This is unlikely to be due to a lack of Gish activity, since it is implicated in several signalling pathways in the wing disc (*Li et al., 2016*; *Gault et al., 2012*). However, CKI isoforms are known to exist in distinct subcellular pools associated with components of the signalling pathways they regulate (*Jiang, 2017*; *Knippschild et al., 2014*). It is therefore possible that recruitment of CkIα to Crb/Ex may involve an adaptor protein expressed in the wing disc, as is the case for CkIε, which requires the DEAD-box

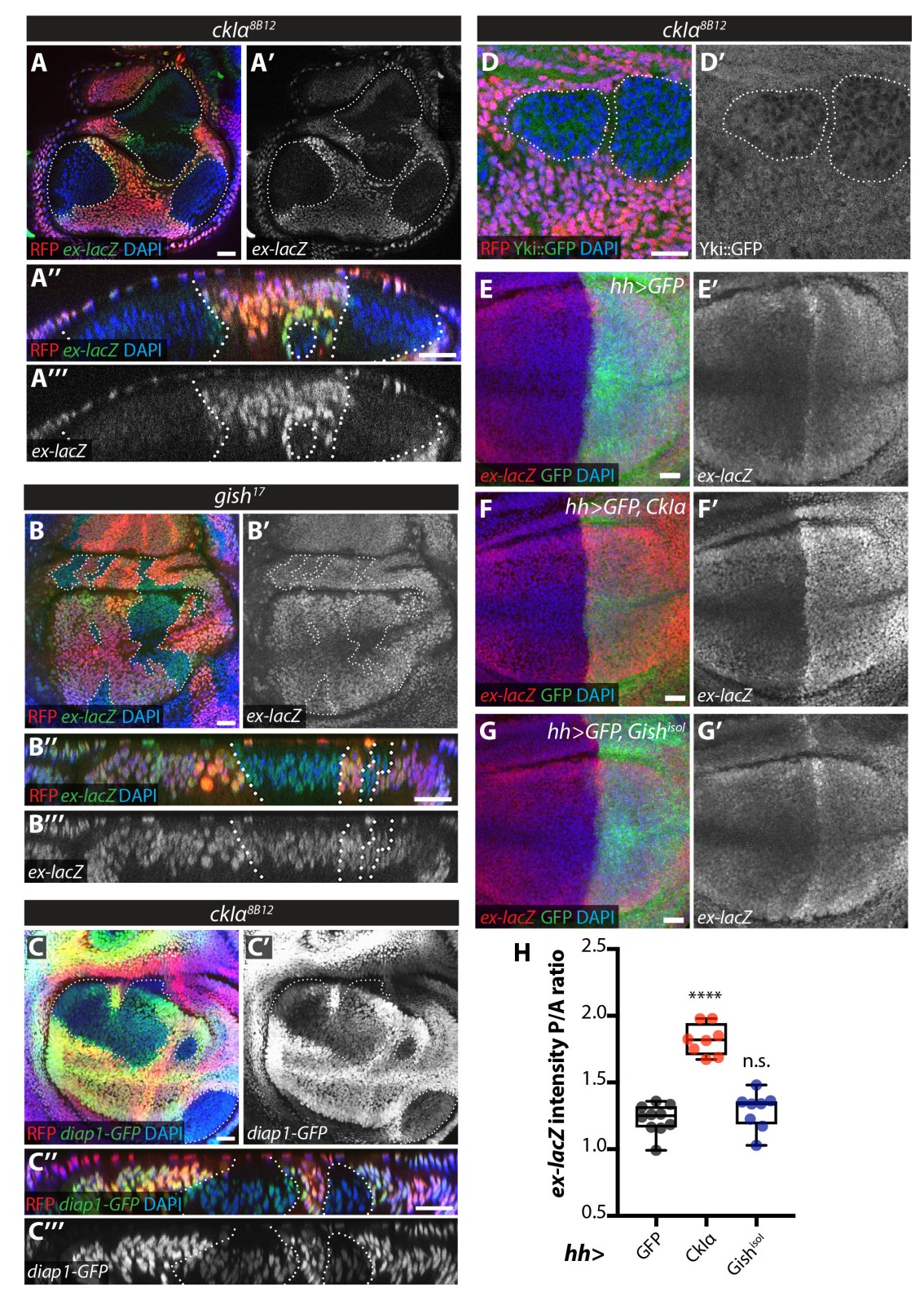

**Figure 7.** Effect of Cklα and Gish loss- and gain-of-function on Yki target gene expression. (**A–D**) Yki transcriptional activity is reduced by loss of *cklα*, but unaffected by loss of *gish*. XY and XZ confocal images of third instar wing imaginal discs bearing clones mutant for *cklα*[8B12] (**A, C, D**), or *gish*[17] (**B**), co-expressing the Yki transcriptional reporter genes *ex-lacZ* (**A, B**), *diap1-GFP3.5* (**C**), or a Yki::GFP fusion protein (a knock-in at the endogenous locus, (**D**). *ex-lacZ* is visualised by immunostaining for β-galactosidase (green in A, A'', B and B'', grey in A', (**A''', B' and B'''**); *diap1-GFP* and Yki::GFP are

*Figure 7 continued on next page*

*Figure 7 continued*

visualised by direct GFP fluorescence (green in C, C'' and D, grey in C', (**C'''** and **D'**). Clones are marked by absence of RFP (red) and highlighted by white dashed lines; DAPI (blue) stains nuclei. Reporter gene expression is drastically reduced in *cklα*[8B12] (**A, C**), but not *gish*[17] (**B**), mutant clones. Yki::GFP appears excluded from the nucleus of *cklα*[8B12] mutant cells (**D**). Scale bars 20 μm. (**E–H**) Overexpression of Cklα, but not Gish[isol], results in upregulation of *ex-lacZ*. Maximum intensity projections of z-stacks of the pouch region of wing imaginal discs from third instar larvae overexpressing no transgene (**E**), *UAS-Cklα* (**F**), or *UAS-Gish*[isol] (**G**) under the control of *hh-Gal4*. Crosses were raised at 25 ℃ and larvae were dissected at wandering L3 stage. *ex-lacZ* expression was detected by immunostaining for β-galactosidase (red in E-G, grey in E'-G'); the posterior compartment is marked by expression of GFP (green); DAPI (blue) stains nuclei. (**H**) Quantification of the posterior to anterior ratio of *ex-lacZ* signal intensity in the pouch region; Cklα expression significantly upregulates this Yki target gene (p=0.0001, one-way ANOVA comparing all means to *hh >control*, with correction for multiple comparisons; n ≥ 8 for all genotypes), while Gish[isol] does not (p=0.4808). Scale bars 20 μm.

DOI: https://doi.org/10.7554/eLife.46592.020

The following source data is available for figure 7:

**Source data 1.** Source data for quantification of relative levels of the Ex::GFPin vivoreporter in wing imaginal discs expressing RNAi targeting CKI kinases alone or in combination with expression of [Crbintra].
DOI: https://doi.org/10.7554/eLife.46592.021

RNA helicase DDX3 to phosphorylate Dishevelled2 in the mammalian Wnt pathway (*Cruciat et al., 2013*).

As Ex apical accumulation is predicted to antagonise Yki, we measured several readouts of Yki-dependent transcription and found that Yki activity is indeed decreased upon *cklα* depletion and increased by Cklα overexpression, while Gish manipulations had no such effect (*Figure 7A–C,E–H*). *cklα* mutations would therefore be predicted to reduce tissue growth by reducing Yki activity. However, we observed that *cklα* mutant clones are generally overgrown, round, and tend to delaminate from the wing disc epithelium, so that they often do not contribute to the adult wing (*Figure 5C* and *Figure 7A,C*). This overgrowth phenotype is due to a strong elevation of Wingless (Wg) signalling in *cklα* mutant tissue (*Legent et al., 2012*) and is very similar to the loss of two Wg antagonists, *axin* and *APC1/2* (Adenomatous Polyposis Coli) (*Muñoz-Descalzo et al., 2011*). Thus, it is likely that the decreased growth occurring as a result of partial loss of Yki activity in *cklα* mutant clones is compensated by a strong increase in Wg signalling.

Our data also revealed that loss of either *cklα* or *slmb* function results in an increase in the apical levels of Crb as well as Ex (*Figure 5E–F* and *Figure 5—figure supplement 1E*). This is not due to a general increase in apical domain size, since aPKC is not affected (*Figure 5—figure supplement 1F–G*). These observations raise two possibilities; either that Crb stability is regulated by the same mechanism as Ex, or that Crb and Ex can affect each other's apical localisation. It seems unlikely that Crb is co-degraded with Ex in a Slmb/β-TrCP-dependent manner, since Crb lacks clear β-TrCP and CKI consensus sequences in its intracellular domain. Moreover, it has been previously shown that Crb levels increase in *ex* mutant clones (*Chen et al., 2010*), a result not easily reconcilable with co-degradation. Instead, we favour the second possibility, in which excess Ex accumulation leads to increased Crb apical levels. Consistent with this idea, over-expression of Ex[1-468] leads to a robust increase in apical Crb (*Figure 5D*). Apical Crb endocytosis occurs through docking of the AP-2 (adaptor-protein 2) complex member α-adaptin, which competes with Stardust to bind the Crb intracellular domain (*Lin et al., 2015*). It is therefore possible that excess Ex interferes with Crb recognition by α-adaptin, thereby decreasing Crb internalisation rates in *cklα* or *slmb* mutant clones.

CRB3, the mammalian orthologue of Crb, can regulate Hippo signalling by mechanisms analogous to those described in *Drosophila*. CRB3 interacts with two proposed orthologues of Ex, Angiomotin (AMOT), and FRMD6, in a density-dependent manner (*Varelas et al., 2010*; *Angus et al., 2012*; *Mao et al., 2018*; *Moleirinho et al., 2013*). It would therefore be interesting to test if the stability of either AMOT or FRMD6 is regulated by CRB3, CKI and β-TrCP. In summary, our data support a model whereby the apical polarity determinant Crb promotes Ex phosphorylation by CKI family kinases, which in turn stimulates Slmb/β-TrCP-mediated ubiquitylation and degradation. We propose that this mechanism facilitates precise and dynamic control of Ex protein levels at the apical membrane, which is crucial for the regulation of Hippo signalling.

## Materials and methods

### *Drosophila* cell culture, expression constructs and chemical treatments

Work involved the use of the *Drosophila* cell line Schneider S2 (RRID:CVCL_Z232). Cells were obtained from the ATCC and mycoplasma testing revealed no contaminations. *Drosophila* S2 cells were grown in *Drosophila* Schneider's medium (Thermo Fisher Scientific) supplemented with 10% (v/v) FBS, 50 µg/mL penicillin and 50 µg/mL streptomycin. Expression plasmids were transfected using Effectene transfection reagent (QIAGEN). Expression plasmids were generated using Gateway technology (Thermo Fisher Scientific). ORFs were PCR amplified from cDNA clones obtained from the *Drosophila* Genomics Resource Center (DGRC, https://dgrc.cgb.indiana.edu/vectors/Overview) and cloned into Entry vectors (pDONR207, pDONR-Zeo). Vectors from the *Drosophila* Gateway Vector Collection and an in-house V5 tag expression vector were used as destination vectors (*Ribeiro et al., 2014*). All Entry vectors were verified by sequencing. Point mutations were generated using the Quikchange Site-Directed Mutagenesis kit (Agilent). The Ex full length, $Ex^{1-468}$ and $Crb^{intra}$ plasmids were previously described (*Ling et al., 2010*; *Genevet et al., 2010*; *Ribeiro et al., 2014*). The WT and mutant versions of Crb differ by specific amino acid substitutions in its intracellular portion ($Y^{10}P^{12}E^{16}$ to $A^{10}A^{12}A^{16}$) (*Ling et al., 2010*). The NTAN-FLAG (N-terminal amidohydrolase 1, an enzyme from the N-end rule pathway that associates with a small number of proteins in *Drosophila* S2 cells), HA-ubiquitin and $Crb^{4A\ mut}$ plasmids were kind gifts from M. Ditzel (University of Edinburgh, Edinburgh), P. Meier (Institute for Cancer Research, London) and B. Thompson (Francis Crick Institute, London), respectively. mScarlet (85042; RRID:Addgene_85042) and pCFD3 (49410; RRID:Addgene_49410) plasmids were obtained from Addgene. Where indicated, proteasome inhibition was achieved by treating cells with 50 µM MG132 (Cambridge Bioscience) and 50 µM calpain inhibitor I (Ac-LLnL-CHO or LLnL) (Sigma) for 4 hr before cell lysis or with 5 µM MG132 overnight.

RNAi production and treatment dsRNAs were synthesised using the Megascript T7 kit (Thermo Fisher Scientific) according to the manufacturer's instructions. DNA templates for dsRNA synthesis were PCR amplified from genomic DNA or plasmids encoding the respective genes using primers containing the 5' T7 RNA polymerase-binding site sequence. dsRNA primers were designed using the DKFZ RNAi design tool (http://www.dkfz.de/signaling2/e-rnai/). The following primers were used: *lacZ* (Fwd –TTGCCGGGAAGCTAGAGTAA and Rev – GCCTTCCTGTTTTTGCTCAC); *gish* (Fwd – TGGCCAAAGAATACATTGATTTAGA and Rev – GGCAGTGAACCCCTTAAGAAATAC); *cklα* (Fwd – GGAGTGCATCAGAGAAGGAGAAC and Rev – GTGGGTGTGTTATGCAAGTATGTT); *ckl$^{pan}$* (Fwd – GAATTAATACGACTCACTATAGGGAGAGGCCATCAAGATGGAGAGC and Rev – GAATTAATACGACTCACTATAGGGAGACATGTAATCTGGCTGCTCC) (*Liu et al., 2002*); *dco* (Fwd – ACGCAGGCATTTAATTCTGTTT and Rev – GGTGTCCTTTGTTTCTTTTACACA); *slmb* (Fwd – TGTACTGTAGGCAGGCGATG and Rev – AGGTGATCATCAGTGGCTCC); *sgg$^{24}$* (Fwd – AGCTCTCAATACAGCCCAGC and Rev – CGGTTCCTGCTGTTGCTC); *sgg$^{25}$* (Fwd – CCGCAATTTCAAAAGAACTC and Rev – AAAATGACAATCGATCAGCG). After cell seeding, S2 cells were incubated with 15–20 µg dsRNA for 1 hr in serum-free medium, before complete medium was added. 72 hr after dsRNA treatment, cells were lysed and processed as detailed below.

### Immunoprecipitation and immunoblot analysis

For purification of FLAG-tagged proteins, cells were lysed in lysis buffer (50 mM Tris pH 7.5, 150 mM NaCl, 1% Triton X-100, 10% (v/v) glycerol, and 1 mM EDTA), to which 0.1M NaF, phosphatase inhibitors 2 and 3 (Sigma) and protease inhibitor cocktail (Complete, Roche) were added. Cell extracts were spun at 17,000 g for 10 min at 4℃. FLAG-tagged proteins were purified using anti-FLAG M2 Affinity agarose gel (Sigma) for >1 hr at 4℃. FLAG immunoprecipitates were then washed four times with lysis buffer before elution using 150 ng/µl 3x FLAG peptide for 15–30 min at 4℃. For isolation of ubiquitylated proteins, cells were collected by centrifugation and washed with cold PBS. 10% of cell material was lysed as described above. The remaining 90% was lysed with boiling 1% SDS-PBS for 5 min. Following quick vortexing, samples were incubated for 5 min at 100℃ before 5-fold dilution using 0.5% BSA-1% Triton X-100-PBS. DNA was sheared by sonication and cell extracts were cleared by centrifugation at 17,000 g for 10 min at 4℃. Samples were diluted 2-fold with 0.5% BSA-1% Triton-X-100-PBS and incubated overnight at 4℃ with monoclonal anti-HA agarose beads (Sigma) using Bio-Spin Columns (Bio-Rad). Following incubation, samples were washed with 0.5%

BSA-1% Triton-X-100-PBS and 1% Triton X-100-PBS before elution. HA immunoprecipitates were eluted from HA beads using 0.2 M glycine pH 2.5 for 30 min at room temperature and eluted samples were equilibrated with 1 M $NH_4HCO_3$. Detection of purified proteins and associated complexes was performed by immunoblot analysis using chemiluminescence (Thermo Fisher Scientific). Western blots were probed with mouse anti-FLAG (M2; Sigma; RRID:AB_262044), mouse anti-Myc (9E10; Santa Cruz Biotechnology; RRID:AB_262044), rat anti-HA (3F10; Roche Applied Science; RRID:AB_2314622), mouse anti-V5 (Thermo Fisher Scientific; RRID:AB_2556564), mouse anti-Crb (Cq4; Developmental Studies Hybridoma Bank, DSHB; RRID:AB_528181), or mouse anti-tubulin (E7; DSHB; RRID:AB_528499). For densitometry analysis of immunoblots, X-ray blots were scanned using an Epson Perfection V700 flatbed scanner and further analysed with the Gel Analyzer function on ImageJ (RRID:SCR_003070). Relative Ex protein levels (normalised to the respective tubulin loading control) were represented as a ratio to the average of the Ex levels in the $Crb^{\Delta FBM}$ negative control samples.

## Immunostaining

Larval tissues were processed as previously described (*Genevet et al., 2010*). Primary antibodies were incubated overnight at 4°C unless otherwise stated. Rat anti-Ci155 antibody (2A1; DSHB; RRID:AB_2109711) was used at 1:50 or 1:100, chicken anti-GFP (ab13970; Abcam; RRID:AB_300798) was used at 1:1000, rat anti-Crb-ICD (a kind gift from F. Pichaud) was used at 1:200, rabbit anti-aPKC (sc-216, Santa Cruz; RRID:AB_2300359) was used at 1:500, and mouse anti-β-galactosidase (Z3781, Promega; RRID:AB_430877) was used at 1:500. Anti-mouse, anti-rat and anti-rabbit Rhodamine Red-X-, FITC-, or Cy5-conjugated (Jackson ImmunoResearch) secondary antibodies were used at 1:400 or 1:500. Anti-rat Alexa Fluor 488-conjugated secondary antibody (Thermo Fisher) was used at 1:500. Anti-rat Alexa Fluor 568-conjugated (Abcam), anti-chicken Alexa Fluor 488-conjugated (Abcam) or anti-mouse Alexa Fluor 647-conjugated (Jackson ImmunoResearch) secondary antibodies were used at 1:1000. Secondary antibodies were incubated for at least 2 hr at room temperature. After washes, tissues were mounted in Vectashield (with or without DAPI) (H-1000 or H-1200, Vector Labs; RRID:AB_2336789 and RRID:AB_2336790, respectively), or stained with DAPI (1 µg/mL) for 10 min before mounting with Mowiol 40–88 (Sigma). Fluorescence images were acquired on Zeiss LSM510 Meta, Zeiss LSM710 or Zeiss LSM880 confocal laser scanning microscopes (40x objective lens).

## Protein sequence alignments

CKI kinases were compiled with BLAST (NCBI; RRID:SCR_004870) using the full-length *Drosophila* CkIα sequence as query. Sequence alignments were performed using MUSCLE (RRID:SCR_011812) (*Edgar, 2004*) and CKI kinase phylogeny was designed using the PhyML (RRID:SCR_014629) and DrawTree tools in the Phylogeny online tool (http://www.phylogeny.fr/simple_phylogeny.cgi) (*Dereeper et al., 2008*). Protein sequence accession numbers were as follows: NP_727631 (*D. melanogaster* CkIα); NP_001020276.1 (*H. sapiens* CKIα); NP_733414 (*D. melanogaster* Dco); NP_001884.2 (*H. sapiens* CKIδ); NP_001885.1 (*H. sapiens* CKIε); NP_001163628 (*D. melanogaster* Gish isoform I); NP_001014628 (*D. melanogaster* Gish isoform F); NP_071331.2 (*H. sapiens* CKIγ isoform 1); NP_001310.3 (*H. sapiens* CKIγ isoform 2); NP_004375.2 (*H. sapiens* CKIγ isoform 3); NP_572794 (*D. melanogaster* CG2577); NP_609851 (*D. melanogaster* CG7094); NP_608697 (*D. melanogaster* CG9962); NP_649536 (*D. melanogaster* CG12147).

## *Drosophila* genetics and genotypes

Transgenic RNAi stocks were obtained from the Vienna *Drosophila* Resource Center (VDRC; RRID:SCR_013805) (*cklα^RNAi^*: 107574KK; *gish^RNAi^*: 108680KK; *dco^RNAi^*: 330069^shRNA^). *ubi-Ex^1-468^::GFP* (wt or S453A) transgenes were cloned using Gateway technology into a modified pKC26-pUbiq plasmid (*Gaspar et al., 2015*). *UAS-CKI* transgenes were cloned using Gateway technology into the pUASg-HA(N)-attB vector. *UAS-Ex^FL^* (wt or S453A) transgenes were cloned into the pUAST-attB plasmid (*Bischof et al., 2007*) and generation of transgenic flies was performed by BestGene (RRID:SCR_012605). *UAS-Ex^1-468^* (wt or S453A) transgenes were cloned into the pUASg-attB plasmid (*Bischof et al., 2013*) and transgenic flies were generated in the Crick Fly Facility. Transgenes were inserted at 62E1 (BL-9748) or 28E7 (BL-9723) using ΦC31-mediated integration. *gish^17^* and *CG7094^F2^* mutants were generated by CRISPR-mediated gene editing. gRNA plasmids were injected

by the Fly Facility of the Department of Genetics, University of Cambridge. $crb^{82-04}$ was obtained from Duojia Pan (UT Southwestern). $slmb^{9H4-17}$ was obtained from Daniel St Johnston (Gurdon Institute, UK). $slmb^1$ was obtained from Daniel Kalderon (Columbia University). $dco^{le88}$ was obtained from Tomas Dolezal (University of South Bohemia). $ckl\alpha^{8B12}$ was obtained from Jessica Treisman (NYU Skirball Institute). $gish^{K03891}$ was obtained from Bloomington (BL-13263).

All crosses were raised at 25°C unless otherwise stated. Genotypes were as follows:

**Figure 3A**, **Figure 3—figure supplement 1A**: hsFLP;; FRT82B ubi-RFP, ubi-Ex$^{1-468}$::GFP/FRT82B $crb^{82-04}$

**Figure 3B**, **Figure 3—figure supplement 1B**, **Figure 5—figure supplement 2A–C**: hsFLP;; FRT82B ubi-RFP, ubi-Ex$^{1-468}$::GFP/FRT82B $slmb^{9H4-17}$

**Figure 3C**: hsFLP/+; ; FRT82B $slmb^{9H4-17}$, $crb^{82-04}$/ubi-Ex$^{1-468}$::GFP, FRT82B ubi-RFP

**Figure 3D, H**: w;; hh-Gal4, ubi-Ex$^{1-468}$::GFP / +

**Figure 3E**, **Figure 3—figure supplement 1C**: w;; hh-Gal4, ubi-Ex$^{1-468}$::GFP/UAS-crb$^{intra}$

**Figure 3F**: w;; hh-Gal4, ubi-Ex$^{1-468\ S453A}$::GFP / +

**Figure 3G**, **Figure 3—figure supplement 1D**: w;; hh-Gal4, ubi-Ex$^{1-468\ S453A}$::GFP/UAS-crb$^{intra}$

**Figure 3I**: w;; hh-Gal4, ubi-Ex$^{1-468}$::GFP/UAS-HA::ckl$\alpha$

**Figure 3J**: w;; hh-Gal4, ubi-Ex$^{1-468}$::GFP/UAS-HA::gish$^{isol}$

**Figure 3—figure supplement 1E**: w; Act > y$^+$>Gal4 / +; hh-Gal4, ubi-Ex$^{1-468}$::GFP/UAS-crb$^{intra}$

**Figure 3—figure supplement 1F**: w; UAS-lacZ$^{RNAi}$ / +; hh-Gal4, UAS-CD8::GFP / +

**Figure 3—figure supplement 1G**: w; UAS-lacZ$^{RNAi}$ / +; hh-Gal4, ubi-Ex$^{1-468}$::GFP / +

**Figure 3—figure supplement 1H**: w; UAS-lacZ$^{RNAi}$ / +; hh-Gal4, ubi-Ex$^{1-468\ S453A}$::GFP / +

**Figure 3—figure supplement 1J**: w; en-Gal4, UAS-CD8::GFP/UAS-CD8::GFP (18°C cross)

**Figure 3—figure supplement 1K**: w; en-Gal4, UAS-CD8::GFP / +; UAS-crb$^{intra}$ / + (18°C cross)

**Figure 3—figure supplement 1L**: w; en-Gal4, UAS-CD8::GFP/UAS Ex$^{WT}$ (18°C cross)

**Figure 3—figure supplement 1M**: w; en-Gal4, UAS-CD8::GFP, UAS-Ex$^{WT}$ / +; UAS-crb$^{intra}$ / + (18°C cross)

**Figure 3—figure supplement 1N**: w; en-Gal4, UAS-CD8::GFP/UAS-Ex$^{S453A}$ (18°C cross)

**Figure 3—figure supplement 1O**: w; en-Gal4, UAS-CD8::GFP, UAS-Ex$^{S453A}$ / +; UAS-crb$^{intra}$ / + (18°C cross)

**Figure 5A**: hsFLP;; FRT82B ubi-RFP, ubi-Ex$^{1-468}$::GFP/FRT82B $dco^{le88}$

**Figure 5B**: hsFLP;; FRT82B ubi-RFP, ubi-Ex$^{1-468}$::GFP/FRT82B $gish^{17}$

**Figure 5C**: hsFLP, FRT19A ubi-RFP/FRT19A $ckl\alpha^{8B12}$;; ubi-Ex$^{1-468}$::GFP / +

**Figure 5D**: w;; hh-Gal4, UAS-CD8::GFP/UAS-Ex$^{1-468}$

**Figure 5E**: hsFLP;; FRT82B ubi-GFP/FRT82B $slmb^1$

**Figure 5F**: hsFLP;; FRT82B ubi-GFP/FRT82B $slmb^{9H4-17}$

**Figure 5—figure supplement 1A**: hsFLP;; FRT82B ubi-RFP, ubi-Ex$^{1-468}$::GFP/FRT82B $gish^{KG03891}$

**Figure 5—figure supplement 1D**: hsFLP; FRT40A ubi-RFP/FRT40A CG7094$^{F2}$; ubi-Ex$^{1-468}$::GFP / +

**Figure 5—figure supplement 1E**: hsFLP, FRT19A ubi-RFP/FRT19A $ckl\alpha^{8B12}$

**Figure 5—figure supplement 1F**:, **Figure 5—figure supplement 2D–F**: hsFLP, FRT19A ubi-RFP/FRT19A $ckl\alpha^{8B12}$;; ubi-Ex$^{1-468}$::GFP / +

**Figure 5—figure supplement 1H**: hsFLP; FRT40A CG7094$^{F2}$; FRT82B $gish^{17}$, $dco^{le88}$/FRT82B ubi-RFP, ubi-Ex$^{1-468}$::GFP

**Figure 5—figure supplement 1I**: hsFLP, FRT19A ubi-RFP/FRT19A $ckl\alpha^{8B12}$;; FRT82B $gish^{17}$, $dco^{le88}$/FRT82B ubi-RFP, ubi-Ex$^{1-468}$::GFP

**Figure 6A, B**: w; UAS-ckl$\alpha^{RNAi}$ (107574KK) / +; hh-Gal4, ubi-Ex$^{1-468}$::GFP/tub-Gal80$^{ts}$

**Figure 6C, D**: w; UAS-ckl$\alpha^{RNAi}$ (107574KK) /+; hh-Gal4, ubi-Ex$^{1-468}$::GFP/UAS crb$^{intra}$, tub-Gal80$^{ts}$

**Figure 6E, F**: w; UAS-gish$^{RNAi}$ (108680KK) / +; hh-Gal4, ubi-Ex$^{1-468}$::GFP/tub-Gal80$^{ts}$

**Figure 6G, H**: w; UAS-gish$^{RNAi}$ (108680KK) / +; hh-Gal4, ubi-Ex$^{1-468}$::GFP/UAS crb$^{intra}$, tub-Gal80$^{ts}$

**Figure 6—figure supplement 1A**: w;; hh-Gal4, ubi-Ex$^{1-468}$::GFP/UAS crb$^{intra}$, tub-Gal80$^{ts}$

**Figure 6—figure supplement 1C**: UAS-dco$^{RNAi}$ (330069$^{shRNA}$) / +; hh-Gal4, ubi-Ex$^{1-468}$::GFP / +

**Figure 6—figure supplement 1D**: UAS-dco$^{RNAi}$ (330069$^{shRNA}$) / +; hh-Gal4, ubi-Ex$^{1-468}$::GFP/UAS-crb$^{intra}$

*Figure 6—figure supplement 1E*: tub-Gal4, hs-FLP, UAS-nucGFP::myc / + or Y; UAS-Crb<sup>intra</sup> / +; FRT82B blank/ubi-Ex<sup>1-468</sup>::mScarlet, FRT82B tub-Gal80

*Figure 6—figure supplement 1F*: tub-Gal4, hs-FLP, UAS-nucGFP::myc / + or Y; UAS-Crb<sup>intra</sup> / +; FRT82B gish<sup>17</sup>/ubi-Ex<sup>1-468</sup>::mScarlet, FRT82B tub-Gal80

*Figure 6—figure supplement 1G*: FRT19A blank/tub-Gal80, hs-FLP, FRT19A; UAS-nls-lacZ, UAS-CD8::GFP / +; ubi-Ex<sup>1-468</sup>::mScarlet, UAS-Crb<sup>intra</sup>/tub Gal4

*Figure 6—figure supplement 1H*: cklα<sup>8B12</sup> FRT19A/tub-Gal80, hs-FLP, FRT19A; UAS-nls-lacZ, UAS-CD8::GFP / +; ubi-Ex<sup>1-468</sup>::mScarlet, UAS-Crb<sup>intra</sup>/tub Gal4

*Figure 7A*: cklα<sup>8B12</sup> FRT19A/hsFLP ubi-RFP FRT19A; ex<sup>697</sup> (ex-lacZ) / +

*Figure 7B*: hsFLP / +; ex<sup>697</sup> (ex-lacZ) / +; FRT82B gish<sup>17</sup>/FRT82B ubi-RFP

*Figure 7C*: cklα<sup>8B12</sup> FRT19A/hsFLP ubi-RFP FRT19A; ; diap1-GFP3.5 / +

*Figure 7D*: cklα<sup>8B12</sup> FRT19A/hsFLP ubi-RFP FRT19A; yki::GFP / +

*Figure 7E*: w;; hh-Gal4, UAS-CD8::GFP / +

*Figure 7F*: w;; hh-Gal4, UAS-CD8::GFP/UAS-cklα

*Figure 7G*: w;; hh-Gal4, UAS-CD8::GFP/UAS-gish<sup>isol</sup>

## Immunofluorescence quantification and statistical analyses

For quantification of Ex<sup>1-468</sup>::GFP (wt or S453A) (*Figure 3J*, *Figure 3—figure supplement 1R*) or Crb (*Figure 3—figure supplement 1Q*) ratios in posterior versus anterior compartment, two transverse sections were acquired per disc (A-P; mid-dorsal and mid-ventral pouch) and the pixel intensity of the GFP (Ex) or Cy5 (Crb) signal along the apical region of the cells was measured in the two compartments using Fiji (RRID:SCR_002285). $n \geq 14$ for all genotypes; one-way ANOVA with Dunnett's multiple comparisons test. Quantification of aPKC levels (*Figure 5—figure supplement 1G*) were performed as above but using transverse sections spanning mutant clones. Data represents 42 clones from 11 wing discs (*Figure 5—figure supplement 1G*). Significance was calculated by unpaired t-test. For *Figure 6—figure supplement 1B*, Ex<sup>1-468</sup>::GFP ratio was calculated as aforementioned and significance was calculated using a two-way ANOVA (comparing all genotypes to Crb<sup>intra</sup> within each timepoint) with Dunnett's multiple comparisons test. For *Figure 7H,* P/A ratio of ex-lacZ intensity was calculated by manually drawing around each compartment of the pouch in maximum intensity projections, then measuring the mean grey pixel value in Fiji. Significance was calculated using a one-way ANOVA comparing all means to *hh >control*, with correction for multiple comparisons; $n \geq 8$ for all genotypes.

## Generation of CRISPR/Cas9 mutants

gRNA sequences were as follows: *gish* AATGGAGCCTATGAAGTCAA; *CG7094*-up ATATCTCGGAC TAAGCATCA; *CG7094*-down ACGGGGTTGTGAGCCTCAGC. gRNA expression plasmids were created by ligation of annealed oligos into pCFD3 (Addgene 49410) (*Port et al., 2014*), diluted to 100 ng/µl and injected into *nos-Cas9 Drosophila* embryos (stock CFD-2; Fly Facility, Department of Genetics, University of Cambridge). Progeny of injected animals were screened for homozygous lethality (*gish*) or PCR screened to identify a deletion (*CG7094*; primers F1 tcgtgtgaacatcgtggtcgt and R2 ctttcggttggcagctttgtc). The *gish<sup>17</sup>* mutation was genotyped using primers Gish_PCR_F1 GCGAATGTGTTGCTTTGGTG and Gish_M17_mut2 GTGTAGTTGCGGAGCCTTTC (318 bp amplicon was obtained specifically from mutant allele).

## Analysis of genetic interactions in *Drosophila* adult wings

For analysis of genetic interactions in the *Drosophila* wing, flies with the genotypes of interest were collected and preserved in 70% EtOH for at least 24 hr. Wings were removed in 100% isopropanol, mounted in microscope slides using Euparal (Anglian Lepidopterist Supplies) as mounting medium and baked at 65°C for at least 5 hr. Adult wing images were captured using a Pannoramic 250 Flash High Throughput Scanner (3DHISTECH) and extracted using the Pannoramic Viewer software (3DHISTECH). Wing area was quantified using ImageJ (the alula and costal cell of the wing were both excluded from the analysis). Images were processed using Adobe Photoshop (RRID:SCR_014199).

## Mass spectrometry analysis

AP-MS experiments followed a GeLC MS/MS approach. Gel lanes were fragmented into eight equally sized pieces and subjected to in-gel trypsin digestion using a Perkin Elmer Janus Automated Workstation. Peptide mixtures were acidified to 0.1% TFA and injected onto a nanoACQUITY UPLC (Waters Corporation) coupled to an LTQ-Orbitap XL (Thermo Fisher Scientific) via an Advion Biosciences Nanomate. Peptides were eluted over a 30 min gradient (5–40% ACN). Mascot distiller was used to extract peak lists, which were searched with Mascot v.2.4.1 (Matrix Science; RRID:SCR_014322) against the *Drosophila melanogaster* Uniprot reference proteome. Methionine oxidation was entered as a variable modification and search tolerances were 5 ppm and 0.8 Da for peptides and fragments, respectively. Individual lane searches were combined and results compiled in Scaffold 4.0.3 (RRID:SCR_014345).

## Acknowledgements

We thank the Vienna *Drosophila* Resource Center for providing transgenic RNAi fly stocks used in this study. Stocks obtained from the Bloomington *Drosophila* Stock Center (NIH P40OD018537) were used in this study. The antibodies Cq4, E7 and 2A1, respectively developed by E Knust, M Klymkowsky and R Holmgren were obtained from the Developmental Studies Hybridoma Bank, created by the NICHD of the NIH and maintained at The University of Iowa, Department of Biology. We thank M Ditzel, P Meier, B Thompson and Addgene (49410 and 85042) for plasmids and F Pichaud for the Crb-ICD antibody. We thank D St Johnston, D Kalderon, J Treisman, D Pan and T Dolezal for providing fly stocks. We thank Linda Hammond and the Crick Advanced Light Microscopy Facility for assistance with microscopy. We thank the Crick Fly Facility (T Gilbank, S Maloney, C Gillen, S Murray, G Davies and J Kurth) for support with fly husbandry and P Faull for help with Mass Spectrometry. We thank the Crick Genomics Equipment Park for assistance. We thank the Fly Facility at the Department of Genetics, University of Cambridge for help with generation of CRISPR mutants. We thank members of the Ribeiro and Tapon labs for helpful discussions and SA Martin, SA Godinho, JF Marshall and H McNeill for critical reading of the manuscript. The authors declare no conflicts of interest. This work was supported by funding from Cancer Research UK (C16420/A18066) and The Academy of Medical Sciences/Wellcome Trust Springboard Award (SBF001/1018). Work in the Tapon lab is supported by the Francis Crick Institute, which receives its core funding from Cancer Research UK (FC001175), the UK Medical Research Council (FC001175), and the Wellcome Trust (FC001175), as well as a Wellcome Trust Investigator award (107885/Z/15/Z).

## Additional information

### Funding

| Funder | Grant reference number | Author |
|---|---|---|
| Cancer Research UK | C16420/A18066 | Alexander D Fulford<br>Paulo S. Ribeiro |
| Academy of Medical Sciences | SBF001/1018 | Alexander D Fulford<br>Paulo S. Ribeiro |
| Wellcome | SBF001/1018 | Alexander D Fulford<br>Paulo S. Ribeiro |
| Cancer Research UK | FC001175 | Maxine V Holder<br>David Frith<br>Ambrosius P Snijders<br>Nicolas Tapon |
| Medical Research Council | FC001175 | Maxine V Holder<br>David Frith<br>Ambrosius P Snijders<br>Nicolas Tapon |
| Wellcome | FC001175 | Maxine V Holder<br>David Frith<br>Ambrosius P Snijders<br>Nicolas Tapon |

| Wellcome | 107885/Z/15/Z | Nicolas Tapon |
| --- | --- | --- |

The funders had no role in study design, data collection and interpretation, or the decision to submit the work for publication.

## Author contributions

Alexander D Fulford, Conceptualization, Data curation, Formal analysis, Validation, Investigation, Visualization, Methodology, Writing—original draft, Writing—review and editing, Designed, performed and analysed in vivo experiments and biochemical experiments, and wrote the manuscript with input from all the authors; Maxine V Holder, Conceptualization, Data curation, Formal analysis, Validation, Investigation, Visualization, Methodology, Writing—original draft, Writing—review and editing, Designed, performed and analysed in vivo experiments and performed CRISPR-mediated gene editing for generation of new gish and CG7094 mutants, and wrote the manuscript with input from all authors; David Frith, Formal analysis, Investigation, Methodology, Writing—review and editing, Performed and analysed mass spectrometry; Ambrosius P Snijders, Formal analysis, Supervision, Investigation, Methodology, Writing—review and editing, Performed and analysed mass spectrometry; Nicolas Tapon, Conceptualization, Supervision, Funding acquisition, Writing—original draft, Project administration, Writing—review and editing, Designed and supervised the project, and wrote the manuscript with input from all authors; Paulo S Ribeiro, Conceptualization, Resources, Data curation, Formal analysis, Supervision, Funding acquisition, Investigation, Visualization, Methodology, Writing—original draft, Project administration, Writing— review and editing, Designed and supervised the project, designed, performed and analysed in vivo and biochemical experiments, and wrote the manuscript with input from all authors

## Author ORCIDs

Alexander D Fulford (iD) https://orcid.org/0000-0002-8880-1720
Maxine V Holder (iD) https://orcid.org/0000-0002-4613-2194
Nicolas Tapon (iD) https://orcid.org/0000-0001-5267-6510
Paulo S Ribeiro (iD) https://orcid.org/0000-0002-6020-6321

## Decision letter and Author response

Decision letter https://doi.org/10.7554/eLife.46592.024
Author response https://doi.org/10.7554/eLife.46592.025

# Additional files

## Supplementary files

• Transparent reporting form
DOI: https://doi.org/10.7554/eLife.46592.022

## Data availability

All data generated or analysed during this study are included in the manuscript and supporting files.

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
