## [Decision Letter]

Thank you for submitting your article "Casein kinase 1 family proteins promote Slimb-dependent expanded degradation" for consideration by *eLife*. Your article has been reviewed by two peer reviewers, and the evaluation has been overseen by a Reviewing Editor and Utpal Banerjee as the Senior Editor. The reviewers have opted to remain anonymous.

The reviewers have discussed the reviews with one another and the Reviewing Editor. They have agreed that, in principle your manuscript is appropriate for *eLife*, but that a number of revisions including new experimental data would be required to bring it to the standards of the journal. The full comments of the reviewers are appended below. Although you are not obligated to provide new data relevant to every point, several important points have been raised that definitely require attention. In particular, point #1 from reviewer 1, and major points 1 and 5 from reviewer 2 (relating to organ growth) require clear conclusions.

Reviewer #1:

The work of Fulford et al. fills the gap in knowledge of the kinase that phosphorylates Ex and targets it for degradation, downstream of Crb overexpression. The authors provide sufficient in vivo and biochemical data to support their conclusions. Overall, the data are of high quality and well controlled; I recommend publication in *eLife* after revisions.

1) My major concern/confusion is the lack of any discussion about the seeming contradiction that both loss of Crb and excess Crb leads to Ex degradation. How can Crb be necessary for Ex stabilization at the membrane and at the same time target it for degradation? What is it about suboptimal Crb levels that triggers Ex degradation? Are Slimb and CK1 proteins required for Ex degradation upon Crb loss as well? The authors could check this by generating double loss-of-function clones for Crb and Slimb or CK1 proteins, and assaying Ex levels.

2) in vivo, at least in wing discs, CK1α seems to be the major kinase. Is this also true in other imaginal discs? They could extend their results to other discs by analyzing *slimb* and CK1α mutant clones with and without Crb overexpression.

3) The authors state: "When a Crbintra transgene was expressed in the posterior compartment using the *hedgehog-Gal4* driver (*hh-Gal4*), ubi-Ex^1-468^::GFP was lost from the apical surface and degraded (Figure 3C-D and Figure 3—figure supplement 1C). In contrast, the S453A mutant reporter accumulated all over the cell upon Crbintra expression, presumably due to the fact that it cannot be recognised by Slmb and degraded (Figure 3E-F and Figure 3—figure supplement 1D). "

I disagree with the conclusion that Ex^S453A^ mutant accumulates all over the cell, it does not accumulate at the apical membrane. Therefore, it seems to be still degraded at the membrane suggesting involvement of other residues. Alternatively, it is mislocalized to the basolateral membrane by ectopic Crb and cannot be degraded there.

Reviewer #2:

It has been demonstrated in previous papers that Crb triggers Ex turnover in a Slmb/β-TrCP-dependent manner. However, recognition by Slmb/β-TrCP requires substrate phosphorylation, which had remained an unexplored aspect of Ex regulation. In this paper, Fulford et al. provide evidence that the CKI family kinases, CkIα and Gish, act downstream of Crb to promote phosphorylation of Ex, in turn promoting Ex degradation. The paper comprises a large amount of supporting data, which is generally of very good quality and consistent with the authors' interpretations. The strong points of this manuscript are the thoroughness of the biochemistry assays although it is still unclear about the phosphorylation site(s) of CKI on Ex. Overall, this paper gave me a feeling that the authors described a quite simple biological context using a very complicated way. I would suggest the authors simplify this paper. In addition, the major weakness of this paper is that it doesn't show any biological function of CKI in regulating organ size and Yki activity, which dramatically reduced the significance of this work. My concerns listed below should be addressed in revision.

1) To assess the functional significance of CKI kinases in controlling Hippo pathway activation, it will be important to examine whether CKI controls organ growth. CkIα (and/or Gish) mutant- or overexpression- MARCM clone assays would be required.

2) Following the above question, the authors need to examine Yki activity in CkIα (and/or Gish) mutant/overexpression background using Diap1-lacZ or *ex-lacZ* reporter.

3) The authors showed that CKI phosphorylates Ex in vitro, however, it is important to identify the phosphorylation site(s) of Ex and generate the phosphor-specific antibody to examine the endogenous Ex phosphorylation. Alternatively, the phosphor-tag gel or pan-phosphor S/T antibody could be used for the phosphorylation detection.

4) All biochemistry assays were using overexpression system. They should detect endogenous Ex expression and degradation. For example, CHX treatment could be used to avoid *gish*-mediated transcriptional regulation of endogenous Ex expression.

5) Although the biochemistry assays have shown that CKI kinase Gish is required for Crb-mediated Ex turnover, the genetic epistasis between Crb and CkIα in controlling Yki activation and organ growth is still needed to be tested. I would suggest the authors perform the following experiments "*hh*> Crbintra vs. *hh*>Crbintra + CkIαRNAi", then compare the Yki reporter induction (e.g. Diap1-LacZ) and P-compartment size.

---

## [Author Response]

Reviewer #1:[…] 1) My major concern/confusion is the lack of any discussion about the seeming contradiction that both loss of Crb and excess Crb leads to Ex degradation. How can Crb be necessary for Ex stabilization at the membrane and at the same time target it for degradation? What is it about suboptimal Crb levels that triggers Ex degradation? Are Slimb and CK1 proteins required for Ex degradation upon Crb loss as well? The authors could check this by generating double loss-of-function clones for Crb and Slimb or CK1 proteins, and assaying Ex levels.

We apologise for not being sufficiently clear in our description of previous data regarding the effect of Crb on Ex protein levels. Previous work from ourselves and others had indeed shown that Crb is required to recruit Ex to the apical plasma membrane (Ling et al., 2010, Chen et al., 2010, Hafezi et al., 2012, Ribeiro et al., 2014). However, as well as promoting Ex activation by membrane recruitment, Crb also brings Ex into proximity with CKI family members and the SCF^Slimb^ complex, as shown by our Affinity Purification/Mass Spectrometry analysis (Figure 1A and Ribeiro et al., 2014). Thus, we believe Crb is both responsible for Ex recruitment to its site of activity and also for ensuring its fast turnover through degradation in order to fine-tune Yki regulation. We have clarified this issue in the first paragraph of the Discussion.

The reviewer raises in interesting point in enquiring about the fate of Ex in the cytoplasm upon loss of Crb. Previously published data indicated that endogenous Ex accumulates in the cytoplasm upon Crb loss (Ribeiro et al., 2014, Robinson et al., 2010, Ling et al., 2010), which had been interpreted as meaning that Ex is mainly degraded at the apical plasma membrane, and failure of apical recruitment leads to stabilisation and cytoplasmic accumulation. However, our Ex::GFP reporter does not accumulate in the cytoplasm in Crb clones (Figure 4A, A’), suggesting a different possibility: that endogenous Ex cytoplasmic accumulation is primarily due to *ex* transcriptional upregulation in *crb* mutant clones. Indeed, work published by the Jiao lab suggests SCF^Slimb^ can also degrade Ex in the cytoplasm (Zhang et al., 2015). To directly test this possibility and as requested by the reviewer, we examined the Ex::GFP sensor in *crb, slmb* double mutant clones (new Figure 3C, C’). We observed that, as anticipated, loss of *slmb* leads to Ex sensor cytoplasmic accumulation in *crb* mutant clones, confirming the finding from the Jiao lab that Ex can be degraded in the cytoplasm and supporting the idea that transcriptional elevation of *ex* plays a role in cytoplasmic accumulation of Ex protein in Crb clones. Thus, although Crb promotes Ex degradation by SCF^Slimb^ at the plasma membrane, there is also a Crb-independent Slimb-dependent cytoplasmic Ex degradation pathway, though the relative turnover rate of these mechanisms is not clear. Since Ex activity in growth control is thought to occur at the apical plasma membrane (Fulford et al., 2018), it is likely that the Crb-dependent mechanism is key in determining Ex functional levels. We discuss this new result in the first paragraph of the Discussion.

2) in vivo, at least in wing discs, CK1α seems to be the major kinase. Is this also true in other imaginal discs? They could extend their results to other discs by analyzing slimb and CK1α mutant clones with and without Crb overexpression.

As requested by the reviewer, we examined the levels of the Ex::GFP reporter in haltere, leg and eye imaginal discs. In the haltere and leg disc, the levels of the Ex reporter were elevated in both *ckI*α and *slmb* clones, suggesting a similar mechanism operates as in the wing disc. Interestingly, we did not see a clear increase in the eye disc. However, *ckI*α and *slmb* loss in these tissues elicited a very strong clone extrusion phenotype, and a complete loss of ommatidial differentiation, making the analysis difficult, especially since the most prominent Ex localisation in the eye disc is at the apical side of the emerging ommatidial clusters. These new data are presented in Figure 5—figure supplement 2.

3) The authors state: "When a Crbintra transgene was expressed in the posterior compartment using the hedgehog-Gal4 driver (hh-Gal4), ubi-Ex^1-468^::GFP was lost from the apical surface and degraded (Figure 3C-D and Figure 3—figure supplement 1C). In contrast, the S453A mutant reporter accumulated all over the cell upon Crbintra expression, presumably due to the fact that it cannot be recognised by Slmb and degraded (Figure 3E-F and Figure 3—figure supplement 1D). "I disagree with the conclusion that Ex-S453A mutant accumulates all over the cell, it does not accumulate at the apical membrane. Therefore, it seems to be still degraded at the membrane suggesting involvement of other residues. Alternatively, it is mislocalized to the basolateral membrane by ectopic Crb and cannot be degraded there.

We agree with the reviewer that it is possible that the Ex^S453^ reporter is mislocalised to the basolateral membrane by ectopic Crb and is refractory to degradation there. We have changed the manuscript text to reflect this possibility. “It is therefore possible that Ex^1-468 S453A^ is mislocalised to the basolateral membrane by ectopic Crb, where it cannot be degraded”.

Reviewer #2:[…] Overall, this paper gave me a feeling that the authors described a quite simple biological context using a very complicated way. I would suggest the authors simplify this paper. In addition, the major weakness of this paper is that it doesn't show any biological function of CKI in regulating organ size and Yki activity, which dramatically reduced the significance of this work. My concerns listed below should be addressed in revision.

We thank the reviewer for the insightful comments and suggestions, which have helped us improve the manuscript. We believe that whilst it is tempting to consider that Expanded regulation by Crb is a simple biological process, the reality is that the existence of multiple layers of regulation and the fact that there is Yki-dependent feedback regulation of transcriptional levels of *ex* make these studies challenging to perform and to explain in a simple manner. Nevertheless, based on the reviewer’s comments, we have endeavoured to highlight the relevance of the biological phenomenon studied and to clarify our arguments in the manuscript accordingly. We believe that our revised manuscript and new data significantly improve the quality and strength of our findings.

1) To assess the functional significance of CKI kinases in controlling Hippo pathway activation, it will be important to examine whether CKI controls organ growth. CkIα (and/or Gish) mutant- or overexpression- MARCM clone assays would be required.2) Following the above question, the authors need to examine Yki activity in CkIα (and/or Gish) mutant/overexpression background using Diap1-lacZ or ex-lacZ reporter.

We agree with the reviewer that it is important to correlate the stabilisation of Ex in *ckI*α mutant clones with Yki activity. To address this, we examined two transcriptional readouts of Yki activity: *ex-lacZ*, an enhancer trap in the *ex* locus (Hamaratoglu et al., 2006), and *diap1-GFP3.5*, a fragment of the *diap1* promoter driving a GFP reporter (Zhang et al., 2008), both of which are widely used in the field. Results of this analysis are shown in the new Figure 7. In accordance with our model, while *gish* mutant clones had wild type levels of *ex-lacZ* (Figure 7B), both *diap1-GFP3.5* and *ex-lacZ* levels were markedly reduced in *ckI*α clones (Figure 7A and 7C). In addition, we used a Yki-GFP knock-in fly line (Fletcher et al., 2018) to show that Yki is excluded from the nucleus in *ckI*α mutant cells compared with control (Figure 7D). Thus, decreased Yki activity is correlated with increased Ex stability upon CkIα disruption, while Gish loss has no effect. We then examined the effect of CKI overexpression on *ex-lacZ*. In further support of our model, CkIα overexpression increased *ex-lacZ* expression, while Gish expression had no effect (Figure 7E-H).

The effect on growth is more complicated to address because, like many kinases, CkIα likely regulates a number of substrates. In fact, rather than being smaller as expected from decreased Yki activity, *ckI*α clones are larger than control clones and are often excluded from the disc epithelium once they reach a certain size. This overgrowth phenotype is due to excess Wingless (Wg) signalling, and can be suppressed by expression of dominant negative TCF (Legent et al., 2012). In fact, the clonal overgrowth and exclusion phenotype of *ckI*α mutants closely resembles that of the Wg signalling negative regulators *axin* and *APC1/2* (see for example Munoz-Descalzo et al., 2011). Thus, it is likely that the undergrowth induced by Ex stability and consequent Yki downregulation is masked by the very strong overgrowth due to Wg signalling upregulation upon *ckI*α loss. We now consider these points in the Discussion. In the case of CkIα overexpression, we do observe a mild overgrowth phenotype (Author response image 1), which correlates with increased Yki activity (as measured by *ex-lacZ* expression, shown in Figure 7). As expected, Gish overexpression causes neither *ex-lacZ* expression (Figure 7) nor overgrowth (Author response image 1). It is possible that this overgrowth is observed because Wg and Hippo signalling are sensitive to a different threshold of CkIα activity, so that mild gain of CkIα increases Yki activity without strongly impairing Wg. However, as it is difficult to verify this interpretation, we have not included the effect of CKI overexpression on wing size in the manuscript, though we are happy to include it if the reviewer so requires.

**Author response image 1. respfig1:** Overexpression of CkIα, but not Gish^isoI^, causes a mild overgrowth of the posterior compartment of the wing. (**A-C**) Adult wings from female flies overexpressing no transgene (**A**), UAS-CkIα (**B**), or UAS-Gish^isoI^ (**C**), in the posterior compartment of the developing wing under the control of *hhGal4*. Crosses were raised at 25 °C. (**D**) Quantification of the ratio of the area of the posterior compartment to total wing area in adult wings. Overexpression of CkIα causes a significant overgrowth (p=0.0001, one way ANOVA comparing all means to *hh*> controls, with correction for multiple comparisons). Gish^isoI^ overexpressing wings tend to be slightly smaller than controls, though this is not statistically significant (p=0.0547), and sometimes exhibit blisters. N≥14 for all genotypes.

3) The authors showed that CKI phosphorylates Ex in vitro, however, it is important to identify the phosphorylation site(s) of Ex and generate the phosphor-specific antibody to examine the endogenous Ex phosphorylation. Alternatively, the phosphor-tag gel or pan-phosphor S/T antibody could be used for the phosphorylation detection.

We performed several experiments to address this point but, unfortunately, we were unable to unequivocally identify the CKI phosphorylation site(s). As suggested by the reviewer, we attempted to raise antibodies that recognise a peptide phosphorylated at S453. We reasoned that since S453 is essential for the Ex:Slmb interaction, it is likely to be phosphorylated in response to Crb^intra^ expression, potentially directly by CKIs. Unfortunately, we were unable to validate the p-S453 antibodies we generated in both rabbits and sheep, in any system we tested (e.g. Western blot; see Author response images 2 and 3) and the antibody failed to recognise Ex. Even in conditions where Ex should be extensively phosphorylated, we failed to detect a band recognising either endogenous or exogenous Ex.

**Author response image 2. respfig2:** Evaluation of rabbit Ex phospho-S453 antibody in *Drosophila* S2 cells. S2 cells were transfected with the indicated plasmids 48h prior to lysis and processing for Western blot analysis. Note that the phospho-S453 Ex antibody does not recognise any protein in the vicinity of the predicted sizes of full-length Ex or Ex^1468^. ø denotes empty vector. GFP and tubulin were used as transfection and protein loading control, respectively.

**Author response image 3. respfig3:** Evaluation of sheep Ex phospho-S453 antibody in *Drosophila* S2 cells. S2 cells were transfected with the indicated plasmids 48h prior to lysis and processing for immunoblotting. Note that the phospho-S453 Ex antibody does not seem to specifically recognise full length Ex or Ex^1-468^. Open and closed circles denote absence or presence of MG132 treatment, respectively. GFP and tubulin were used as transfection and protein loading control, respectively.

We also attempted to monitor the levels of endogenous Ex in *Drosophila* S2 cells. However, we found that the widely used anti-Ex antibody was not sufficiently specific (see Author response image 4). Even though we can detect a band that is likely to represent endogenous Ex, this is extremely weak and would not allow us to generate conclusive data.

**Author response image 4. respfig4:** Evaluation of Ex antibody in the recognition of endogenous Ex protein. S2 cells were transfected with the indicated plasmids 48h prior to lysis and dsRNA treatment was performed 24h before transfection. Note that there are several bands recognised by the Ex antibody. One of the weaker slow migrating bands is consistent with it being endogenous Ex since it is increased when exogenous Ex was transfected and when *wts* was depleted by RNAi. ø denotes empty vector; * indicate non-specific bands recognised by the Ex antibody. Tubulin was used as loading control.

As suggested by the reviewer, we also used Phos-tag gels to examine point mutants of Ex^1-468^ phosphorylatable residues, some of which we had previously shown to be involved in the interaction with Slmb/bTrCP (Ribeiro et al., 2014). Shown in Author response image 5 are the results of this analysis, where Ex^1-468^ was coexpressed with either WT or mutant Crb^intra^ and the lysates were assessed by Phos-tag gel electrophoresis followed by Western blot analysis.

Co-expression of Ex^1-468^ with Crb^intra^ WT, but not DFBM, resulted in a shift in Ex mobility in Phos-tag gels, consistent with our previous results on standard PAGE gels. Our analysis of the Ex mutants confirmed our previous observations regarding their effect on the stability of Ex and their ability to interact with Slmb/b-TrCP. We have previously shown that S453 is crucial for Slmb/b-TrCP recognition and Ex degradation. When S453A is mutated, Ex fails to bind to Slmb/b-TrCP and is therefore more stable. We had previously observed that S453A mutation does not abrogate the Ex mobility shift and this is confirmed with the Phos-tag analysis. Ex^1-468^ is still shifted and it is significantly more stable. Interestingly, unmodified Ex S453A is also seen in the Phos-tag gel, perhaps indicating that, indeed, this residue is phosphorylated in response to expression of Crb. However, it is clear that even if that is the case, it is not the only phosphorylation event triggered by Crb expression.

The other mutations tested (S457A and S462A) behaved very similarly to WT Ex, showing only a slight increase in Ex stability. In the case of S457A, the Ex detected is nearly completely modified, suggesting that this residue may not be phosphorylated in response to Crb expression. In the case of S462A, we could detect some unmodified Ex, again perhaps suggesting that it may be phosphorylated but, like S453, it is not the only residue involved. What is clear from the modest band shifts we observed is that Phos-tag gels do not allow us to readily separate different individual phosphorylated species of Ex any better than standard PAGE gels. Therefore, given the fact that the sequence surrounding the Slmb/b-TrCP consensus site is extremely rich in Ser and Thr residues, this approach is very unlikely to allow us to identify the (potentially multiple) CKI phosphorylation sites.

**Author response image 5. respfig5:** Evaluation of Ex electrophoretic mobility shift using Phos-tag gels. S2 cells were transfected with the indicated plasmids 48h prior to lysis. Lysates were run in Phos-tag gels to assess phosphorylation pattern of Ex^1-468^ (WT, S453A, S457A or S462A) in the presence of Crb^intra^ WT or a variant carrying a mutated FERM-binding motif (DFBM). Immunoblot analysis was performed using the indicated antibodies. Open circle indicates empty vector. Tubulin was used as loading control.

Finally, we attempted to immunoprecipitate CkIα from S2 cell lysates to perform in vitro kinase assays on bacterially expressed Ex followed by Mass Spectrometry analysis. Unfortunately, we could not find conditions where the purified CkIα was active in the in vitro setting and had no time to extensively troubleshoot the experiment given the time constraints on the revision.

4) All biochemistry assays were using overexpression system. They should detect endogenous Ex expression and degradation. For example, CHX treatment could be used to avoid gish-mediated transcriptional regulation of endogenous Ex expression.

As mentioned above, we attempted to monitor the levels of endogenous Ex in *Drosophila* S2 cells. However, we found that the Ex antibody was not sufficiently specific (see Author response image 4) and, unfortunately, this precludes the analysis suggested by the reviewer. As shown in vivo and mentioned by the reviewer, the use of endogenous Ex is complicated by transcriptional feedback, making the use of CHX, which has pleiotropic effects on many signalling pathways, necessary. We therefore believe that use of exogenous Ex allows us to dissect Ex stability in a cleaner experimental setup.

5) Although the biochemistry assays have shown that CKI kinase Gish is required for Crb-mediated Ex turnover, the genetic epistasis between Crb and CkIα in controlling Yki activation and organ growth is still needed to be tested. I would suggest the authors perform the following experiments "hh> Crbintra vs. hh>Crbintra + CkIαRNAi", then compare the Yki reporter induction (e.g. Diap1-LacZ) and P-compartment size.

As requested by the reviewer, we examined the effect of CkIα depletion on the Crb^intra^ phenotype (overgrowth and Yki activity using *ex-lacZ*) in the posterior compartment of the wing disc (Author response image 6). We observed that, consistent with our model, both the overgrowth phenotype and the *ex-lacZ* increase elicited by Crb^intra^ expression are suppressed by CkIα depletion. However, the discs combining Crb^intra^ and CkIα depletion appear heavily disrupted and, therefore, we want to be cautious about the interpretation of this experiment. In particular, it is possible that the suppressed growth is due to toxicity of combining Crb^intra^ with CkIα depletion rather than specifically lowering Yki activity. Nevertheless, we are happy to include the data as supplementary material if required by the reviewer.

**Author response image 6. respfig6:** *ckIα^RNAi^* partially reverts Crb^intra^-induced upregulation of Yki target genes and overgrowth. (A-E) *ckIα^RNAi^* partially reverts Crb^intra^-induced upregulation of the Yki activity reporter *ex-lacZ*. Sum slices projections of z-stacks of the pouch region of third instar wing imaginal discs expressing *ex-lacZ* and either no transgene (**A**), *UAS-ckIα^RNAi^* (**B**), *UAS-Crb^intra^* (**C**), or *UASckIα^RNAi^*plus *UAS-Crb^intra^* (**D**). Transgene expression was driven by *hh-Gal4*, and the time of onset of expression was controlled using *tub-Gal80^ts^*; crosses were raised at 25 °C then shifted to 29 °C 48 h prior to dissection as wandering L3 larvae. The posterior compartment is marked by CD8::GFP (green); DAPI (blue) stains nuclei. Scale bars 20 μm. (**A’-D’**) *ex-lacZ* channels alone. (**E**) Quantification of posterior to anterior ratio of nuclear *ex-lacZ* intensity in sum slices projections. Expression of Crb^intra^ for 48 h induces a significant increase in *ex-lacZ* expression levels, which is partially reverted by co-expression of *ckIα^RNAi^* (p=0.7240, one way ANOVA comparing all means to *hh>GFP* control, with correction for multiple comparisons, n≥11 for all genotypes) (F-I) *ckIα^RNAi^* partially reverts Crb^intra^-induced overgrowth of wing imaginal discs. XY sections of whole wing imaginal discs from third instar larvae, which have expressed the indicated transgenes under the control of *hh-Gal4* for 48 h prior to dissection. The posterior compartment is marked by CD8::GFP (green); DAPI (blue) stained nuclei. *hh>Crb^intra^* discs tend to be larger overall, and the posterior compartment of the pouch develops ectopic folds as the disc attempts to accommodate the additional cells. Coexpression of *ckIα^RNAi^* results in less severe overgrowth, and less pronounced ectopic folding of the pouch region. Scale bars 100 μm. Genotypes for Rebuttal Figure 8: (A, A’, F) *w; ex-lacZ/+; hh-Gal4, UAS-CD8::GFP/+*; (B, B’, G) *w; UAS-ckIα^RNAi^ (110768KK)/ ex-lacZ; hh-Gal4, UAS-CD8::GFP/ tub-Gal80^ts^*; (C, C’, H) *w; ex-lacZ/+; hh-Gal4, UAS-CD8::GFP/ UAS-Crb^intra^, tub-Gal80^ts^*; (D, D’, I) *w; UASckIα^RNAi^ (110768KK)/ ex-lacZ; hh-Gal4, UAS-CD8::GFP/ UAS-Crb^intra^, tub-Gal80^ts^*